# A dopamine-gated learning circuit underpins reproductive state-dependent odor preference in *Drosophila* females

Ariane C Boehm[1,2], Anja B Friedrich[1], Sydney Hunt[1], Paul Bandow[1,2,3], KP Siju[1], Jean Francois De Backer[1], Julia Claussen[1], Marie Helen Link[1], Thomas F Hofmann[3,4], Corinna Dawid[3,4], Ilona C Grunwald Kadow[1,2,3,5]*

[1]Technical University Munich, School of Life Sciences, Neuronal Control of Metabolism, Freising, Germany; [2]Graduate School of Systemic Neurosciences, Ludwig Maximilian University, Martinsried, Germany; [3]ZIEL – Institute for Food and Health, Technical University Munich, School of Life Sciences, Freising, Germany; [4]Technical University Munich, School of Life Sciences, Chair of Food Chemistry and Molecular Sensory Science, Freising, Germany; [5]University of Bonn, Faculty of Medicine, Institute of Physiology II, Bonn, Germany

*For correspondence:
ilona.grunwald@ukbonn.de

Competing interest: The authors declare that no competing interests exist.

**Abstract** Motherhood induces a drastic, sometimes long-lasting, change in internal state and behavior in many female animals. How a change in reproductive state or the discrete event of mating modulates specific female behaviors is still incompletely understood. Using calcium imaging of the whole brain of *Drosophila* females, we find that mating does not induce a global change in brain activity. Instead, mating modulates the pheromone response of dopaminergic neurons innervating the fly's learning and memory center, the mushroom body (MB). Using the mating-induced increased attraction to the odor of important nutrients, polyamines, we show that disruption of the female fly's ability to smell, for instance the pheromone cVA, during mating leads to a reduction in polyamine preference for days later indicating that the odor environment at mating lastingly influences female perception and choice behavior. Moreover, dopaminergic neurons including innervation of the β'1 compartment are sufficient to induce the lasting behavioral increase in polyamine preference. We further show that MB output neurons (MBON) of the β'1 compartment are activated by pheromone odor and their activity during mating bidirectionally modulates preference behavior in mated and virgin females. Their activity is not required, however, for the expression of polyamine attraction. Instead, inhibition of another type of MBON innervating the β'2 compartment enables expression of high odor attraction. In addition, the response of a lateral horn (LH) neuron, AD1b2, which output is required for the expression of polyamine attraction, shows a modulated polyamine response after mating. Taken together, our data in the fly suggests that mating-related sensory experience regulates female odor perception and expression of choice behavior through a dopamine-gated learning circuit.

## Editor's evaluation

In this manuscript, the authors explore the circuit mechanism underlying mating-induced change of odor preference in *Drosophila*. Olfactory cues during mating induce a long-lasting increase in attraction to polyamines in female flies. The authors use a combination of neurogenetics, imaging, and behaviour to identify elements of the mushroom body and lateral horn circuitry involved in this behaviour. The importance of mushroom body plasticity in female postmating changes highlights a novel pathway for these changes and reveals the variety of mechanisms by which the brain can

encode experience and adapt behavior, making this paper of interest to scientists within the field of reproductive behaviors and neuroscience of internal states.

## Introduction

External stimuli and internal states guide decisions by influencing interpretation of the environment. These decisions not only impact on the individual itself but may have severe effects on its offspring. A female animal, therefore, adapts her choices to her reproductive state. This includes not only the decision to mate, but also a preference for certain, nutrient-rich food sources or, in case of egg-laying animals, oviposition sites (*Chapman and Wolfner, 2017*; Sayin, *Sayin et al., 2018*). Choices pertaining to food, nutrients, or egg-laying sites frequently depend on chemosensory cues such as odors or tastes. In line with this, changes in olfactory and gustatory perception have been reported for pregnant women (*Ochsenbein-Kölble et al., 2007*). The cellular and neural mechanisms responsible for such reproductive state-dependent preferences remain incompletely understood.

A powerful mechanism for adapting preference for an odor or taste according to internal state acts within the very sensory neurons that detect the chemical (*Leinwand and Chalasani, 2011*). For instance, hunger increases the sensitivity of sweet taste and quenches that of bitter taste neurons in different animal species including humans (*Carnell et al., 2012*; *Chu et al., 2014*; *LeDue et al., 2016*; *Palouzier-Paulignan et al., 2012*; *Rolls, 2007*). The estrous cycle influences how a female mouse perceives a putative mate (*Dey et al., 2015*). Outside of estrus, the hormone progesterone strongly inhibits the female's male pheromone-sensitive olfactory sensory neurons (OSNs), and thereby, completely blunts her interest in males. Moreover, previous results using volume measurements in human females before, during, and after pregnancy suggested that pregnancy induces long-lasting, global changes in grey matter volume in several brain regions (*Hoekzema et al., 2017*), indicative of modulation in multiple higher brain regions. In the genetically tractable insect, *Drosophila melanogaster*, reproductive state also induces dramatic shifts in the female's chemosensory perception and choices (*Gou et al., 2014*; *Hussain et al., 2016a*; *Ribeiro and Dickson, 2010*; *Walker et al., 2015*). We have previously shown that a mating-induced transient neuropeptidergic modulation of OSNs strongly increases a female fly's preference for higher concentrations of polyamines, a nutrient that significantly increases female egg production (*Hussain et al., 2016a*; *Hussain et al., 2016b*). This rather short-lasting sensory modulation (~24 hr) leads to a longer lasting change in the female's choice behavior, indicating that mating might modulate higher chemosensory processing in the female's central brain. Here, we aimed at investigating the nature and circuit mechanisms of these mating-induced longer lasting behavioral changes.

We used reproductive state-dependent behavior of flies to polyamines to address the more general question of how mating lastingly changes odor perception. Polyamines, namely putrescine, spermine, and spermidine, play essential and conserved roles in most eukaryotic cells and organisms, ranging from DNA replication, cell proliferation, embryonic development to healthy aging (*Miller-Fleming et al., 2015*). High concentrations of putrescine in food increase egg laying in *Drosophila* females (*Hussain et al., 2016b*). More importantly, olfactory detection of polyamines is relatively well characterized at the sensory neuron level and depends on two co-expressed ionotropic receptors (IR), IR76b and IR41a on the fly's antenna (*Hussain et al., 2016b*; *Silbering et al., 2011*). Within the first several hours after mating, the expression of the neuropeptide receptor, sex peptide receptor (SPR), increases by tenfold in the OSNs leading, when bound to its ligand myoinhibitory peptide (MIP), to a depression of polyamine OSN presynaptic output to the second order neurons of the olfactory system (*Hussain et al., 2016a*). While virgin flies show a high attraction toward low concentration of polyamines, mated flies are attracted to higher concentrations (*Hussain et al., 2016a*).

OSNs and second order neurons of the olfactory system, the projection neurons (PNs), synapse in glomeruli in the antennal lobe (AL). PNs pass on the olfactory information to two higher order brain centers of the insect, the mushroom body (MB) and the lateral horn (LH). Output neurons of these brain centers, MB output neurons (MBON) and LH output neurons (LHON), respectively, are thought to pass on this information to neurons eventually projecting to motor neurons (*Bates et al., 2020*; *Schlegel et al., 2021*). Both brain centers also receive input by neuromodulatory neurons. In particular, the role of dopaminergic neurons (DANs) innervating the 15 compartments of the MB, where they form neuromodulatory synapses with MBONs and the MB intrinsic neurons, the Kenyon

cells (KC), have been studied extensively for their role in aversive and appetitive memory formation (*Cognigni et al., 2018*; *Owald and Waddell, 2015*; *Siju et al., 2021*; *Thum and Gerber, 2019*). In a simple model, DANs respond to contextual information such as reward or punishment and thereby modulate the synapse between a specific MBON, usually of the same compartment as the DAN, and the KCs. Most studies have reported a depression of the KC-MBON synapse as a result of context or experience (*Cohn et al., 2015*; *Lewis et al., 2015*; *Owald et al., 2015*). Further evidence indicates that DANs might not only modulate MBONs but also activate them directly (*Takemura et al., 2017*). Interestingly, MB output can affect LH output. Dolan et al. recently reported that learning modulates the response of specific LHONs through an MBON connecting the MB to the LH (*Dolan et al., 2018*).

Attraction to nutrients, egg-laying or nest building sites are considered innate behaviors in most female animals (*Cury et al., 2019*). Innate olfactory behavior is thought to rely on stereotyped circuits connecting specific glomeruli in the AL to specific neurons in the LH (*Chin et al., 2018*; *Ebrahim et al., 2015*; *Min et al., 2013*). In line with this, previous work in the fly has implicated the LH in the control of reproductive behaviors, such as responses to sex pheromones (*Jefferis et al., 2007*; *Ruta et al., 2010*). Neurons innervating or providing output of the LH show largely stereotypic responses to groups of odors (; *Jeanne et al., 2018*). Moreover, recent studies implicated neurons in the LH in innate valence decisions and odor classification (*Dolan et al., 2019*; *Jeanne et al., 2018*; *Strutz et al., 2014*). Interestingly, modulation of innate behavior, for instance in the context of hunger, also involves the MB (*Bräcker et al., 2013*; *Cohn et al., 2015*; *Grunwald Kadow, 2018*; *Heisenberg, 2003*; *Krashes et al., 2009*; *Lewis et al., 2015*; *Sayin et al., 2018*; *Tsao et al., 2018*). In reproductive behavior, dopaminergic neurons (DANs) innervating the MB regulate the mated female's decision of where to lay her eggs (*Azanchi et al., 2013*). DAN odor responses are modulated by the mating state of the female (*Siju et al., 2020*). In addition, it is worth noting that a crucial role for dopamine and the MB have been shown in the mating-induced changes in male courtship learning (*Keleman et al., 2012*; *Siegel and Hall, 1979*).

Here, we have investigated the neural circuits underpinning female reproductive state-dependent odor attraction on the example of polyamine sensing using brain-wide imaging, connectomics, and behavioral analysis. Our combined data suggest that specific dopaminergic neurons relay the experience of mating to the female's MB, thereby inducing a long-lasting change in female olfactory preference.

## Results

### Mating-related signals induce a lasting increase in female polyamine attraction

Mated female flies show an increased attraction to polyamine odors, such as putrescine, as compared to virgin females due to SPR-mediated neuromodulation of polyamine sensitive OSNs (*Hussain et al., 2016a*). The increased attraction, interestingly, lasts for at least 9 days after mating (*Figure 1A, B*), while the neuromodulation of the OSNs appears to decline already several hours after mating (*Hussain et al., 2016a*). We wondered how mating induces such a long-lasting change in olfactory preference in female flies.

It is possible that the same type of neuromodulation and lasting change also affects other olfactory behavior and odors. However, the attraction to the odor of ammonia, another odor released during decomposition of organic matter, or the food odor vinegar did not increase significantly after mating (*Figure 1—figure supplement 1A*,B). Nevertheless, it is conceivable that other odor responses are similarly affected by mating. Moreover, the context or test assay likely also influences the expression of olfactory behavior (e.g. T-maze vs. egg-laying assay).

During copulation, the male fly transfers sperm and seminal fluid into the female reproductive tract. Among the factors transferred with the sperm is one of the ligands of SPR, sex peptide (SP). SP binds to neurons expressing SPR in the female's reproductive tract and induces the so-called canonical post-mating switch, a suite of behaviors associated with reproduction such as increase in egg laying and rejection of males attempting to mate (*Kubli, 2003*; *Yapici et al., 2008*). Although SPR is required for the change in the mated female's polyamine preference behavior, this appears to rely mainly on the other ligands, myoinhibitory peptides (MIPs) (*Hussain et al., 2016a*; *Kim et al., 2010*) as females mated to SP mutant males still undergo the change in polyamine preference behavior

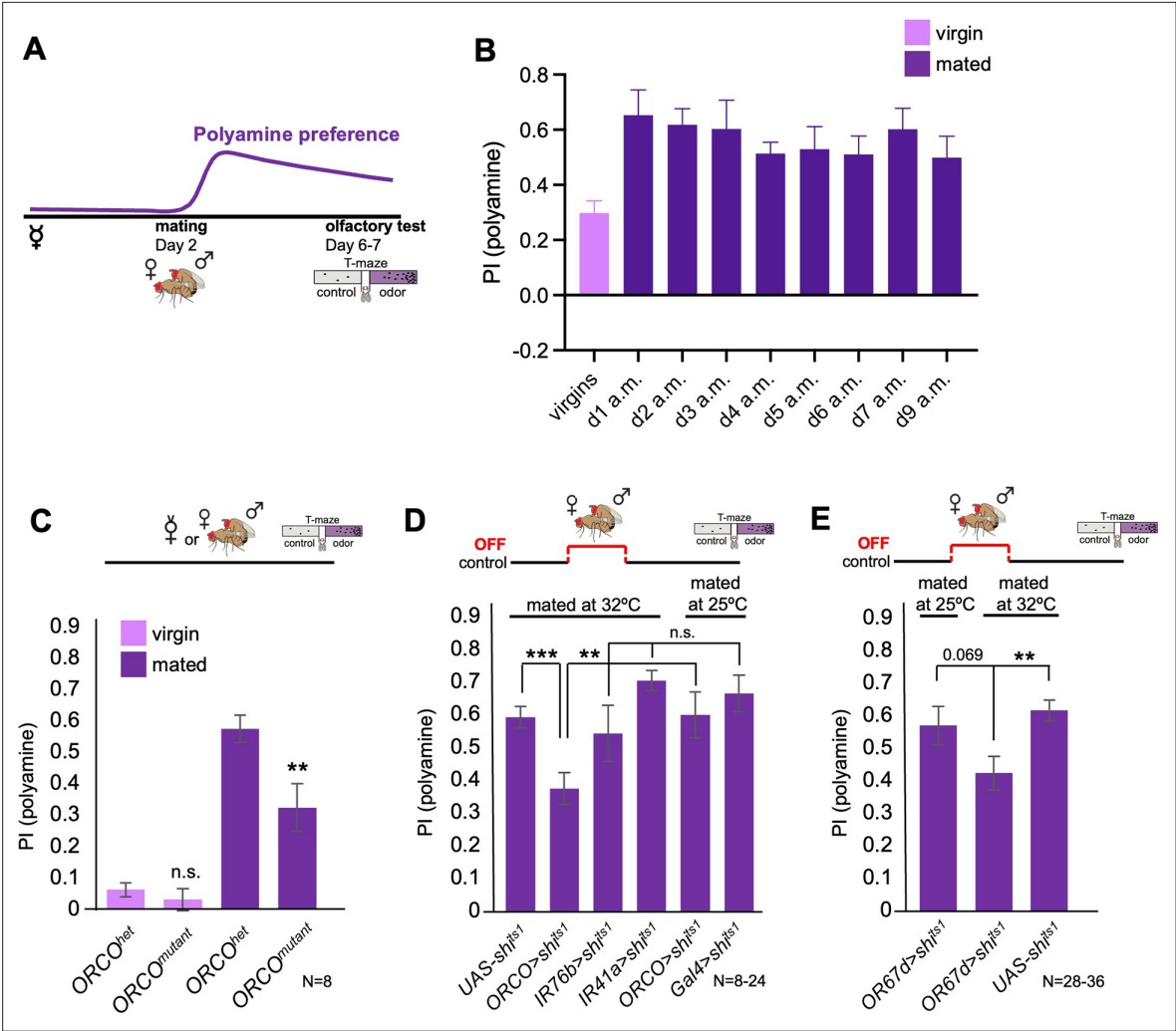

**Figure 1.** Odor experience at mating changes female attraction to polyamines. (**A**) Scheme depicting paradigm used in all experiments. Virgin females were kept alone for 6–7 days or mated on day 2 for 24 hr, then separated, and test at day 6–7 for their preference for the odor of the polyamine, putrescine. Mated females show a high preference for polyamine, which depends on mating-induced modulation of polyamine-detecting olfactory neurons (see text for details). (**B**) CantonS females before mating (virgins) and 1–9 days after mating (a.m.) were analyzed for their preference to putrescine odor as groups in a T-maze. (**C**) Females mutant for ORCO (*ORCO^mutant*) showed a significantly reduced preference for polyamines during olfactory preference test compared to controls (*ORCO^het*). Note that ORCO is not required for the detection of polyamine odor, but for the detection of pheromones and other food related odors. T-maze group assay. Bars depict preference index (PI) ± SEM. Student-s t-test; p-values: n.s.:>0.05, *:<0.05, **:<0.01 (**D**) T-maze group assay for females mated at restrictive temperature (32 °C) and tested at permissive temperature for their polyamine preference (25 °C). In addition, ORCO>shi^ts1 and pooled IR76b and IR41a-Gal4>shi^ts1 females were mated and tested at 25 °C as a control. Note that synaptic inhibition of ORCO-expressing OSNs during mating significantly reduced the females' attraction to putrescine odor. T-maze group assay. (**E**) T-maze group assay for females mated at restrictive temperature (32 °C) or at permissive temperature (25 °C) and tested at permissive temperature for their polyamine preference (25 °C). Note that increased temperature at test leads to a slightly higher attraction to polyamine as compared to testing at 25 °C. This might explain why the difference between flies of the same genotype tested at different temperatures is reduced by not significantly. Bars depict preference index (PI) ± SEM. Student-s t-test; p-values: n.s.:>0.05, *:<0.05, **:<0.01. N>8 groups of 20–40 female flies.

The online version of this article includes the following figure supplement(s) for figure 1:

**Figure supplement 1.** Analysis of the role of polyamines and mating-related cues.

(*Hussain et al., 2016a*). Therefore, we next tested females mated to sterile males. There are two previously used experimental methods to achieve such an effect: males that do not produce either seminal fluid protein or sperm (*Boswell and Mahowald, 1985*; *Chow et al., 2015*). To generate males incapable of producing seminal fluid protein, accessory glands were disrupted through induction of ER stress (see methods; *prd-GAL4;UAS-BiP-RNAi*). After copulation with a seminal fluid protein deficient male, females showed a normal increase in preference for polyamine odor as compared to

controls (*Figure 1—figure supplement 1C*). Similarly, copulation with sperm-deficient males, which were generated by using the male progeny of female mutants of *tudor* (*Chapman et al., 2003*), also led to mated females that behaved indistinguishable from females that mated with wildtype males (*Figure 1—figure supplement 1D*). This result was in line with our previous finding (*Hussain et al., 2016a*), and suggested that not only was SP not essential, but also that other sperm- and seminal fluid-associated factors were dispensable. Moreover, we had previously tested females that, similarly to females mated to sterile males, were unable to produce eggs by generating *ovoD* mutant females. *OvoD* females mated with wildtype males underwent the switch in behavior and were attracted to polyamines to the same extent as controls (*Hussain et al., 2016a*).

We continued to look for additional triggers that could work in synergy with courtship such as signals transferred or present during mating. Male flies perform a complex courtship before attempting to mount the female in order to mate. During this time, females will, among other signals, also smell male-emitted pheromones such as 11-cis-Vaccenyl acetate (cVA). We hypothesized that mating, or more specifically pheromone, could act as a contextual sensory signal able to lastingly modulate higher olfactory processing, perhaps analogous to other sensory contextual experiences such as sweet taste or pain. Our earlier work using in vivo two-photon imaging revealed that some subsets of DANs, presumably primarily through input from KCs, responded strongly to cVA in a mating state-dependent manner suggesting (a) that cVA could act as a contextual signal and (b) that mating induces a lasting change cVA-induced DAN activity. Primarily, DANs innervating a specific compartment of the MB, β'1, responded significantly stronger to cVA in females days after mating as compared to virgins (*Siju et al., 2020*). Interestingly, cVA and other pheromones (e.g. 7-tricosene) are transferred onto the female during courtship and mating, and remain on her body for an extended period of time (~24 hr) (*Ejima et al., 2007*). To test the role of courtship and its signals *vs.* actual mating, we allowed fly couples to go through the courtship ritual but separated them just as the male attempted to mount the female. We then analyzed the odor preference behavior of these females that were technically still virgins. The behavior of these females lay between virgins and mated females and was neither significantly different from virgins nor from mated females consistent with the hypothesis that courtship related signals might contribute as behavioral change triggers (*Figure 1—figure supplement 1E*). Based on this observation and our data above, we next tested one specific aspect of the courtship ritual, odor detection. Pheromone detection relies on the general OR co-receptor, ORCO, and different specific ORs. We thus analyzed mated females mutant for *ORCO*, and hence unable to smell pheromones among other odors. Importantly, ORCO is not required for the detection of polyamine odor, which is mediated by IR76b and IR41a (*Hussain et al., 2016b*; *Silbering et al., 2011*). Mated *ORCO* mutant females displayed a significantly lower polyamine preference compared to controls (*Figure 1C*).

Previous work showed that food odors also enhance courtship and mating (*Grosjean et al., 2011*), and that cVA promotes aggregation behavior of flies on food patches (*Das et al., 2017*; *Lebreton et al., 2015*). Interestingly, polyamines are present in high concentrations in standard fly food (in μg/100 g fly food): histamine (5.49), ethanolamine (676.65), phenylethylamine (5.16), putrescine (172.50), β-alanine (930.84), tyramine (166.64), spermidine (18120.77), and spermine (492.81) (*Figure 1—figure supplement 1F*). Thus, the total amount of polyamines in standard fly food is ~20 mg in 100 g food, which is comparable to levels in very polyamine-rich foods such as oranges (*Kalač, 2014*). To test whether the presence of odors such as pheromones or polyamines was important during mating to induce a higher polyamine odor preference upon mating, we blocked the synaptic output of ORCO and IR76b/IR41a-expressing OSNs exclusively during the 24 hr at and around mating (*Figure 1D*). While synaptic inhibition of IR41a- and IR76b-expressing OSNs during mating had no effect on the mated female's odor preference, blocking the synaptic output of ORCO-OSNs during mating but not at test significantly reduced those females' attraction to the putrescine odor (*Figure 1D*). Similarly, inhibition of synaptic output of cVA-responsive OR67d OSNs during mating but not during test reduced the mated female's attraction to putrescine (*Figure 1E*). Thus, we analyzed whether the smell of cVA was sufficient to induce mated-like female polyamine preference by exposing virgin females in food vials for 24 hr to cVA (50 μl pure solution) or just water on a filter paper. We detected no significant difference between the groups indicating that cVA alone is insufficient to induce mated female odor preference (*Figure 1—figure supplement 1G*).

Next, to test whether the context of food was necessary for females to change their preference upon mating, we removed virgin females for 24 hr from fly food and transferred them to a pure agarose

substrate. We mated 50% of these females to males on agarose, while the other 50% remained alone – hence, they remained virgin – for the same amount of time. After these 24 hr, flies were transferred again onto regular fly food. Virgins behaved as expected and showed no polyamine preference (*Figure 1—figure supplement 1H*). Females mated on agarose showed no significant decrease in polyamine preference compared to females mated on regular fly food (*Figure 1—figure supplement 1H*). We next raised and mated females on a holidic diet containing precise amounts of nutrients such as amino acids but no added polyamines (*Piper et al., 2014*). We again found no difference in the preference for polyamine odor between the two groups of mated females (*Figure 1—figure supplement 1I*).

These results provide evidence that courtship promoting signals such as the pheromone cVA contribute to a switch in preference behavior during mating, but that the presence of food odor, including polyamines, during mating is not required.

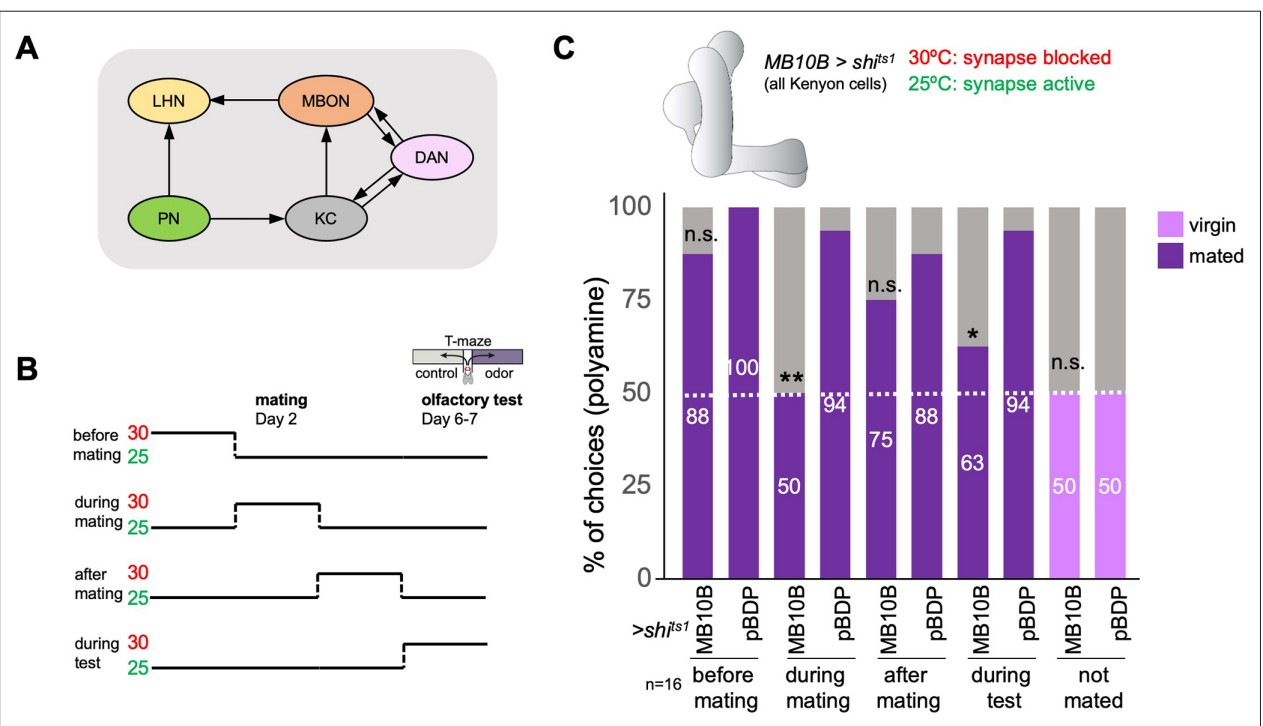

**Figure 2.** The mushroom body and learning mechanisms are required in long-term preference change upon mating. (**A**) Simplified schematic overview of the observed connections between neurons in the olfactory system. PN: projection neuron; LHN: lateral horn neuron; KC: Kenyon cell; DAN: dopaminergic neuron; MBON: mushroom body output neuron. Olfactory sensory neurons connect to PNs in the antennal lobe. PNs project to the two higher olfactory center, the mushroom body (MB) and the lateral horn (LH). In the MB, they provide input to KCs. In the LH, they connect to various types of LHNs. KCs form excitatory synapses onto MBONs. DANs form modulatory synapses onto KCs and MBONs. Note the existence of several recurrent connections. Selected MBONs also project to the LH and make connections with LHNs. This canonical motif can be found in all 15 MB compartments along the MB lobes. Apart from these synaptic connections, additional connections have been shown or proposed. See text for more details. (**B**) Scheme of experimental paradigm used to block Kenyon cell (KC) synaptic output at precise times during the experiment. Shibire$^{ts1}$ was expressed under the control of MB10B-Gal4 in all Kenyon cells (KCs). Temperature shifts from 25 to 30°C at four different time points were used to inhibit KC output before, during, after mating and at test, respectively. At days 6–7 after mating, females were given the choice between the odor of water (control) and the odor of putrescine in a T-maze assay. (**C**) KC synaptic output is required during mating and during preference test. Inhibition of KC output in virgin females during test did not affect putrescine odor preference. Single female preference for putrescine in T-maze assay. 50% represents chance. Fisher's exact test. All p-values depict the comparison between the respective control (*pBDP-Gal4;UAS-shi$^{ts1}$*) and the text group (*MB10B-Gal4;UAS-shi$^{ts1}$*). p-values: n.s.:>0.1, *:<0.1, **:<0.05, ***:<0.001. n=16 single female flies.

The online version of this article includes the following figure supplement(s) for figure 2:

**Figure supplement 1.** Whole brain imaging of polyamine odor responses in mated and virgin flies.

## Mushroom body Kenyon cells promote mating state-dependent odor attraction

Given the role of the MB in olfactory processing and behavioral adaptations and our data above indicating a role of olfaction at the time of mating (see *Figure 1F*), we wondered whether the MB was involved in the mating-induced increase in odor preference. We started with the principal cells of the MB, the KCs, which receive olfactory input through projection neurons (*Figure 2A*). Their long axons form the MB lobes that are innervated by DANs and provide output to MBONs. In addition, they also form recurrent connections with the DANs of the same and in rarer cases additional MB compartments (*Figure 2A*; *Li et al., 2020b*). Given the role of olfaction in courtship and mating, we switched to single fly assays to ensure that mated females were indeed mated; all females were analyzed for successful fertilization upon mating and females that did not produce offspring were discarded from the analysis. We used a set of split-Gal4 lines, generated to target individual cell types of the MB network (*Aso et al., 2014a*). We expressed the temperature-sensitive mutant of dynamin, Shibire$^{ts1}$ (*Kitamoto, 2001*), in all KCs (*MB10B-Gal4;UAS-shits1*), inhibited synaptic transmission by shifting the females to 30°C, and tested their individual polyamine odor preference in the T-maze assay (*Figure 2B*). KC output could be required for the expression of polyamine odor attraction but also for the increase from virgin to mated female behavior. To test this hypothesis, we shifted the females (*MB10B-Gal4;UAS-shi$^{ts1}$*) from 25 to 30°C for 24 hr at four different time points: (1) 24 hr before mating, (2) 24 hr during mating, (3) 24 hr after mating, and (4) at test (*Figure 2A*). Blocking synaptic output during 24 hr before or between mating and test had no significant effect on the mated female's preference for polyamines showing that long-term synaptic inhibition of KCs per se does not affect this behavior (*Figure 2C*, 50% represents chance). By contrast, blocking KC output exclusively during the 24 hr during and around mating, completely prevented the change from virgin to mated female putrescine attraction (*Figure 2C*). Inhibition of KC output during the choice test also reduced the percentage of single females choosing putrescine over water (*Figure 2B and C*), suggesting an involvement of KC output for the expression of polyamine preference. KC output did not affect the choice of virgins (*Figure 2C*). Thus, KC synaptic output appears to be required at two timepoints: First, it is necessary to induce the change in behavior upon mating, and second, it influences the expression of the actual behavioral choice.

Together, these data suggest that MB KCs are required at two time points: for the mating-induced change in female choice behavior and for the expression of this induced choice. Given the important role of the MB in adaptive behavior, it is possible that other (olfactory) behaviors that are dependent on mating state rely on the same or similar mechanism.

## Specific dopaminergic neurons can replace mating experience

Our data suggested that mating-related olfactory signals such as cVA are required to induce increased polyamine attraction after mating. Moreover, MB β'1-innervating DANs respond to cVA in a mating state-dependent manner. We next asked whether these state-dependent changes in odor response where specific to some neurons or whether mating would induce a more wide-spread or even global change in brain activity. For instance, recent evidence suggests that changes in hormone levels trigger circuit maturation in *Drosophila* flies during the first week of life (*Leinwand and Scott, 2021*). We used whole brain lightfield imaging to record the activity of all neurons in the brain in living female flies (*nsyb-Gal4;UAS-CaMP*) (*Aimon et al., 2022*; *Aimon et al., 2019*; *Woller et al., 2021 Figure 2—figure supplement 1A*). Mating did not appear to change brain activity globally in 1-week-old mated and virgin females stimulated with putrescine, and we did not find any significant brain-wide mating state-dependent changes (*Figure 2—figure supplement 1B, C*). Similarly, we did not detect any significant differences between mated and virgin flies by analyzing individual brain regions (*Figure 2—figure supplement 1D, E*). These data suggested that mating does not induce a significant global change in brain state (e.g. general arousal state), but rather relies on modulation of specific neurons and neural circuit elements relevant to a specific task.

Therefore, we sought to test experimentally whether DANs, in particular the DANs responsive to cVA, can induce the switch from virgin to mated female odor choice behavior. We started with line MB188B, because it labeled several PAM DANs with axons in the β'1 lobe, namely PAM-β'1ap and PAM-β'1 m, as well as PAM-γ3, and PAM-γ4 (*Aso et al., 2014a*). We manipulated the activity of these PAMs in virgin and mated females as a replacement for actual mating or during mating (*Figure 3*).

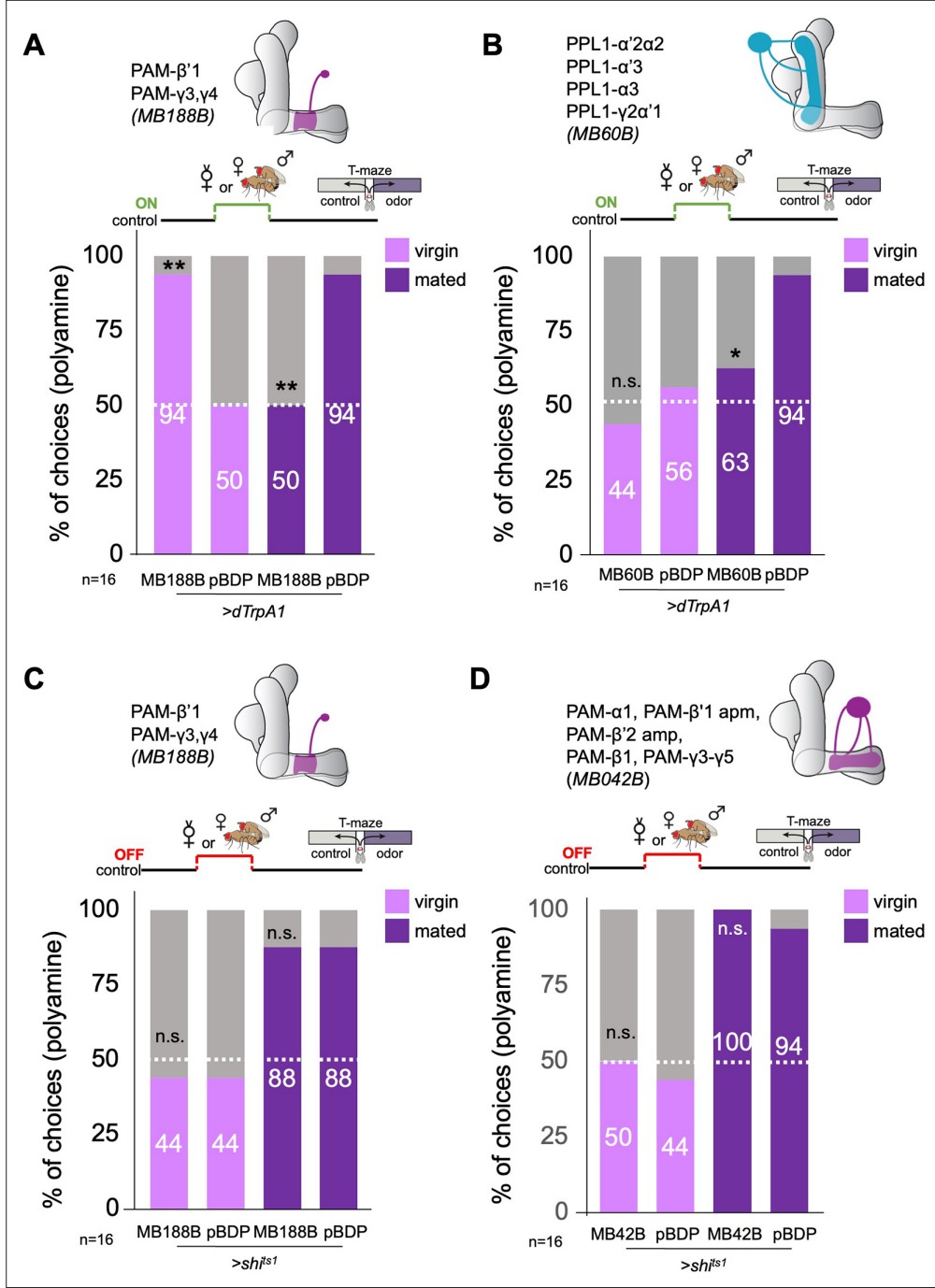

**Figure 3.** β'1 dopaminergic neuron activation can replace mating to induce higher polyamine preference.
(**A**) Activation of PAM-β'1 output (*MB188B-Gal4;UAS-dTrpA1*) instead of mating significantly increased virgin female preference for putrescine at the time of test. Vice versa, it significantly decreased the mated female's preference for the same at testing. (**B**) Activation of PPL1-α'2α2, α3, γ2α'1 output (*MB60B-Gal4;UAS-dTrpA1*) instead of mating also significantly decrease mated female preference for putrescine at the time of test. Note that virgin behavior was not affected. Single female preference for putrescine in T-maze assay. 50% represents chance. (**C**) Inhibition of PAM-β'1 dopaminergic neuron output (*MB188B-Gal4;UAS-shi^{ts1}*) during or instead of mating had no significant effect on mated or virgin female preference for putrescine at the time of test, respectively. (**D**) Inhibition of a large subset of PAM dopaminergic neuron output (*MB042B-Gal4;UAS-shi^{ts1}*) during or instead of mating had no significant effect on mated or virgin female preference for putrescine at the time of test, respectively. Single female preference for putrescine in T-maze assay. 50% represents chance. Fisher's exact test;

*Figure 3 continued on next page*

*Figure 3 continued*

All p-values depict the comparison between the respective control (*pBDP-Gal4;UAS-xx*) and the text group (*MBx-Gal4;UAS-xx*). p-values: n.s.:>0.1, *:<0.1, **:<0.05, ***:<0.001. n=16 single female flies.

The online version of this article includes the following figure supplement(s) for figure 3:

**Figure supplement 1.** Role of PAM dopaminergic neurons during preference test.

**Figure supplement 2.** PAM and PPL dopaminergic neurons during mating or at choice.

Given that β'1 DANs responded to cVA, we again used a single fly T-maze to be able to check every single female fly after an experiment for its mating state (i.e. eggs laid or not). Neuronal activation of these DANs with TrpA1 was highly efficient in switching virgin female preference to mated female preference levels for polyamine (*Figure 3A*). By contrast, the same paradigm with another line, MB109B that expresses in a set of β'2 DANs, did not induce changes in mated or virgin fly behavior (*Figure 3—figure supplement 1D*). Activation of most of the second cluster of DANs innervating the MB, the PPL1-DANs (MB60B) instead of mating, only induced a mild reduction in mated female preference, but had no effect on virgin female behavior (*Figure 3B*). Surprisingly, activation of the same DANs during mating significantly reduced the mated female's attraction to polyamine (*Figure 3A*). Therefore, it appears that β'1,γ3/4 DANs can bidirectionally modulate female choice behavior. This bidirectionality could possibly be achieved through alternative signaling pathways downstream of dopamine receptors as shown for associative learning (*Handler et al., 2019*).

Inhibition of synaptic output of the β'1,γ3/4 DANs during mating had not effect on polyamine preference (*Figure 3C*) indicating that these DANs are sufficient but not strictly necessary for the preference increase. We, therefore, inhibited a large subset of PAM-DANs during mating through expression of shi<sup>ts1</sup> under the control of MB042B (*Figure 3D*). This manipulation did also not affect the polyamine preference of mated females at choice test (*Figure 3D*). Thus, another mechanism, for instance mediated by SPR signaling, compensates for the loss of DAN output. We have previously shown that *SPR* mutants do not show an increased preference for polyamine after mating (*Hussain et al., 2016a*).

We next tested the involvement of DANs in the expression of polyamine preference at the time of T-maze test. Thus, we expressed Shi<sup>ts1</sup> in larger and smaller subsets of PAM- or PPL1-DANs using lines MB42B, MB316B, and MB60B, respectively. Inhibition of PAM-DAN synaptic output at the time of test had no effect on virgin behavior, but significantly reduced mated female attraction to putrescine (*Figure 3—figure supplement 1A, B* and *Figure 3—figure supplement 2A-D*). By contrast, inhibition of PPL1-DAN output resulted in an opposite phenotype; mated females were still highly attracted to the odor, but now also virgins showed a higher attraction to putrescine as compared to controls (*Figure 3—figure supplement 2E-H*). These results are consistent with the interpretation that DANs regulate the expression of female odor attraction in a mating state-dependent manner; although PAM-DANs appear to be necessary for the expression of attraction of mated females, the activity of PPL1-DANs in virgins seems to suppress the expression of odor attraction.

The present data show that specific DANs can mimic mating experience in inducing a change in odor choice behavior. First, DANs innervating the β'1,γ3/4 compartment of the MB bidirectionally modulate odor attraction at the time of mating. Second, additional PAM-DANs including β'2 (see *Figure 3—figure supplement 1A, B*) promote the expression of mated female odor attraction, while PPL1-DANs repress this expression in virgins (see *Figure 3—figure supplement 2E-H*). As for the involvement of KCs, we speculate that this effect is not limited to a mating-induced change in polyamine odor preference but could extend to other odors and possibly additional internal states.

## Specific MB output neurons are involved at different time points during reproductive state-dependent female decision-making

MBONs, as major output of MB KCs and modulatory targets of DANs, are involved in guiding valence-based action selection. Exogenous activation through optogenetics of selected MBONs elicits attraction or aversion, respectively, to light (*Aso et al., 2014b*). In classical associative learning paradigms, DAN activation can replace aversive or appetitive unconditioned stimulus (US) (*Owald and Waddell, 2015*).

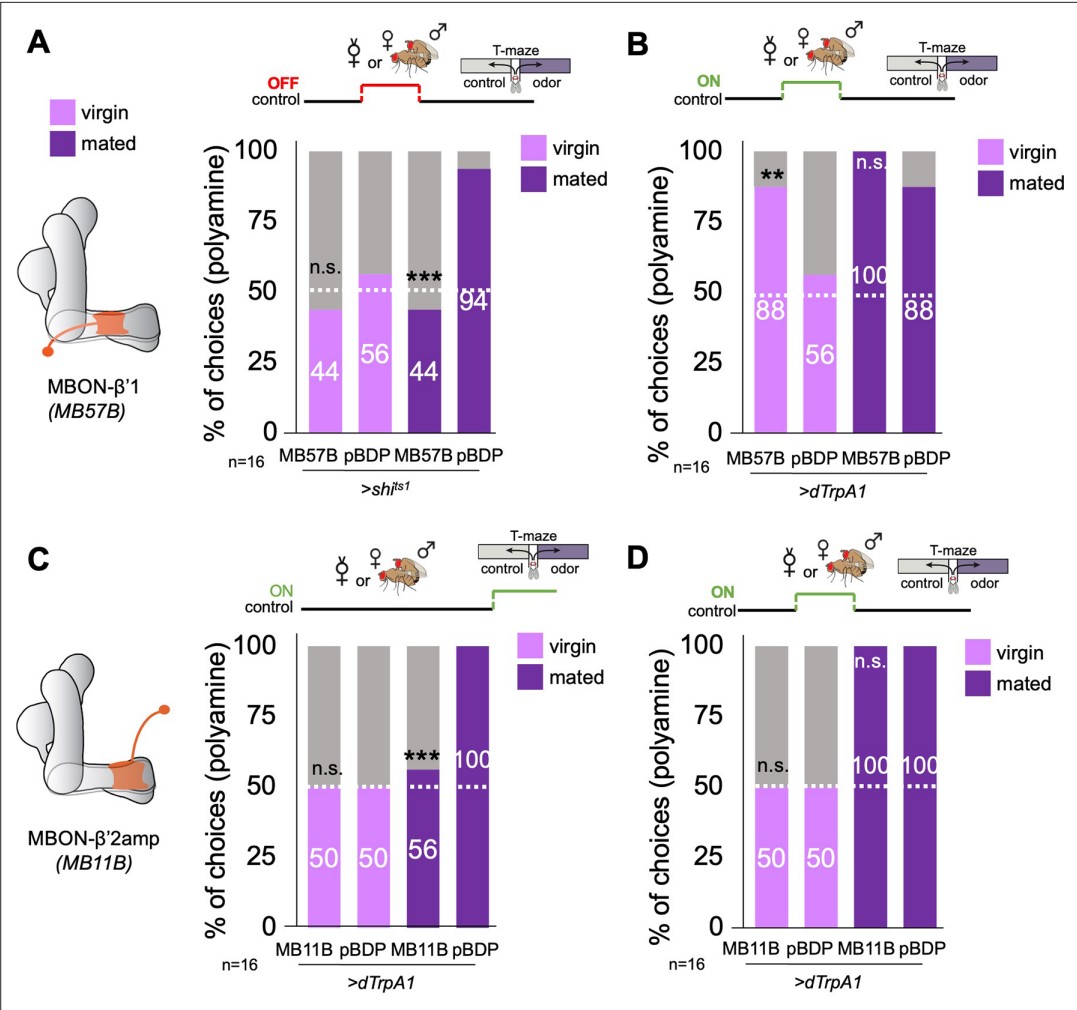

**Figure 4.** β'1 mushroom body output at mating regulates polyamine preference. (**A**) Inhibition of MBON-β'1 output (*MB57B-Gal4;UAS-shi^{ts1}*) during mating significantly reduced mated female preference for putrescine at the time of test. Single female preference for putrescine in T-maze assay. 50% represents chance. (**B**) Activation MBON-β'1 output (*MB57B-Gal4;UAS-dTrpA1*) instead of mating significantly increased virgin female preference for putrescine at the time of test. (**C**) Activation of MBON-β'2 output (*MB11B-Gal4;UAS-dTrpA1*) during test significantly reduced mated female preference for putrescine. (**D**) Activation MBON-β'2 output (*MB11B-Gal4;UAS-dTrpA1*) instead of or during mating did not change female preference for putrescine at the time of test. Single female preference for putrescine in T-maze assay. 50% represents chance. Fisher's exact test; All p-values depict the comparison between the respective control (*pBDP-Gal4;UAS-xx*) and the text group (*MBx-Gal4;UAS-xx*). p-values: n.s.:>0.1, *:<0.1, **:<0.05, ***:<0.001. n=16 single female flies.

The online version of this article includes the following figure supplement(s) for figure 4:

**Figure supplement 1.** Mushroom body output neurons are involved in mating state-dependent choice behavior.

**Figure supplement 2.** Mushroom body output neuron activity influences choice.

---

Given our β'1-DAN imaging data and the effect of DAN activation in lieu of mating, we analyzed the role of MBON-β'1 (aka MBON10) (*Li et al., 2020b*). Blocking of synaptic output of MBON-β'1 during the 24 hr period around mating resulted in a strong reduction of polyamine odor preference in mated females at the time of odor test (*Figure 4A*). suggesting that this MBON's synaptic output critically contributes to the induction of mating-related behavioral changes. In line with this interpretation, dTrpA1-mediated activation of MBON-β'1 in virgins in place of mating significantly increased these females' polyamine preference at the time of choice (*Figure 4B*). Similar to mating which did not modulate ammonia preference (see *Figure 1—figure supplement 1A*), the same manipulation did not increase the virgins' preference for ammonia (*Figure 4—figure supplement 1A*). Interestingly,

activation of MBON-β'1 in virgin females at the time of choice also led to a significant increase in polyamine attraction compared to controls (*Figure 4—figure supplement 1B*). This activation also reduced mated female attraction (*Figure 4—figure supplement 1B*) suggesting a bidirectional role for this MBON at expression of choice. This was similar to the bidirectionality of the DANs innervating the same compartment during mating (see *Figure 3*). In contrast to the PAM-β'1,γ3/4, however, output of MBON-β'1 was required during mating to induce an increased attraction to putrescine (compare *Figures 3C and 4A*). Inhibition of MBON-β'1 output via shibire[ts1] at the time of olfactory test, by contrast, did not significantly alter the mated or virgin female's preference (*Figure 4—figure supplements 1C and 2A*). Together, these results provide evidence that, at the time of mating, MBON-β'1 and DANs innervating the same compartment modulate the degree of polyamine attraction.

In contrast to other MBONs, relatively little is known regarding the function of MBON-β'1 that would explain the observed behavioral phenotypes. MBON-β'1 releases GABA as neurotransmitter and might, therefore, inhibit downstream neuronal activity (*Aso et al., 2014a*). We used the trans-TANGO system and EM connectomics to identify the putatively inhibited downstream neuron (*Figure 4—figure supplement 1E-G*). Expression of trans-TANGO under the control of the MBON-β'1 Gal4-driver primarily and repeatedly labeled a neuron with neurites in the β'2 region of the MB (*Figure 4—figure supplement 1F, F'*). We identified a candidate neuron for our TANGO result using the EM connectome (*Figure 4—figure supplement 1E, G*; *Li et al., 2020b*). PAM-β'2 m (aka PAM06) receives direct input from MBON-β'1. PAM-β'2 m in turn synapses onto MBONs providing output from the β'2 MB compartment (i.e. β'2mp, β'2mp_bilateral; *Figure 4—figure supplement 1G*). Interestingly, β'2 MBONs (i.e. MBON- γ5β'2 a) form direct input synapses with MBON-β'1 (*Figure 4—figure supplement 1G*) suggesting a recurrent loop between the β'1, β'2 MBONs and DANs (*Li et al., 2020b*). Such a recurrent loop could drive persistent behavioral changes as previously shown for male courtship memory (*Zhao et al., 2018*).

MBONs providing output of β'2 were shown to be required for hunger state-dependent innate odor aversion (*Lewis et al., 2015*; *Owald et al., 2015*). Moreover, activation of these MBONs using optogenetics is sufficient to elicit strong avoidance (*Aso et al., 2014b*; *Lewis et al., 2015*). To probe the involvement of β'2 MBONs in mating state-dependent attraction, we used line MB11B-Gal4, which drives Gal4 expression in three different MBON-types, namely MBON-β'2mp, MBON-β'2mp_bilateral, and MBONγ5β'2 a. Indeed, temporary thermogenetic activation of these MBONs (*MB11B-Gal4;UAS-dTrpA1*) at the time of olfactory choice, significantly reduced the mated female's attraction to polyamine (*Figure 4D*). Temporary synaptic output inhibition at the time of choice in virgins increased their attraction to polyamine odor but did not affect the high attraction of mated females (*Figure 4—figure supplement 2B*). By contrast, neither activation nor synaptic output inhibition at the time or in place of mating influenced the female's preference behavior (*Figure 4E*, *Figure 4—figure supplement 1D*). These data suggest that β'2 output is important for the expression of the choice in mated females, but in contrast to β'1 output, it is insufficient to induce the mating-induced change in choice behavior.

We thus propose a similar scenario as observed for learning with the experience of mating providing a contextual signal conveyed to the MB primarily through the β'1 compartment. The β'2 compartment, like in expression of learned behavior, is only necessary to express attraction at the time the mating-induced preference is needed. Given that MBONs are typically not specific to any one odor (*Hige et al., 2015*), we speculate that this mechanism might also affect other mating state-dependent odor perceptions.

## Specific lateral horn output neurons mediate attraction to polyamines

Given the previously reported role of the LH in the expression of learned behavior (*Dolan et al., 2018*), we also analyzed candidate neurons in this brain structure for their role in the expression of polyamine attraction in mated females. Projection neurons (PNs) project from the antennal lobe (AL) glomerulus (VC5), where they receive input by the OSNs detecting polyamine odors (i.e. IR41a/IR76b OSNs) to the LH and the MB calyx (*Figure 5—figure supplement 1A-B'*). As the number of LH output neurons (LHONs) is large, we used the lightfield imaging data (see *Figure 2—figure supplement 1*) to narrow down the putative anatomy of candidate neurons (*Figure 5A*, *nsyb-Gal4;UAS-GCaMP6m*). Stimulation of mated females with putrescine elicited a clear increase in GCaMP fluorescence mainly in the ventral, but also in a smaller area in the dorsal, region of the LH (*Figure 5A*). By contrast, stimulation with 1% vinegar activated primarily neurons in the dorsal part of the LH (*Figure 5A*) as

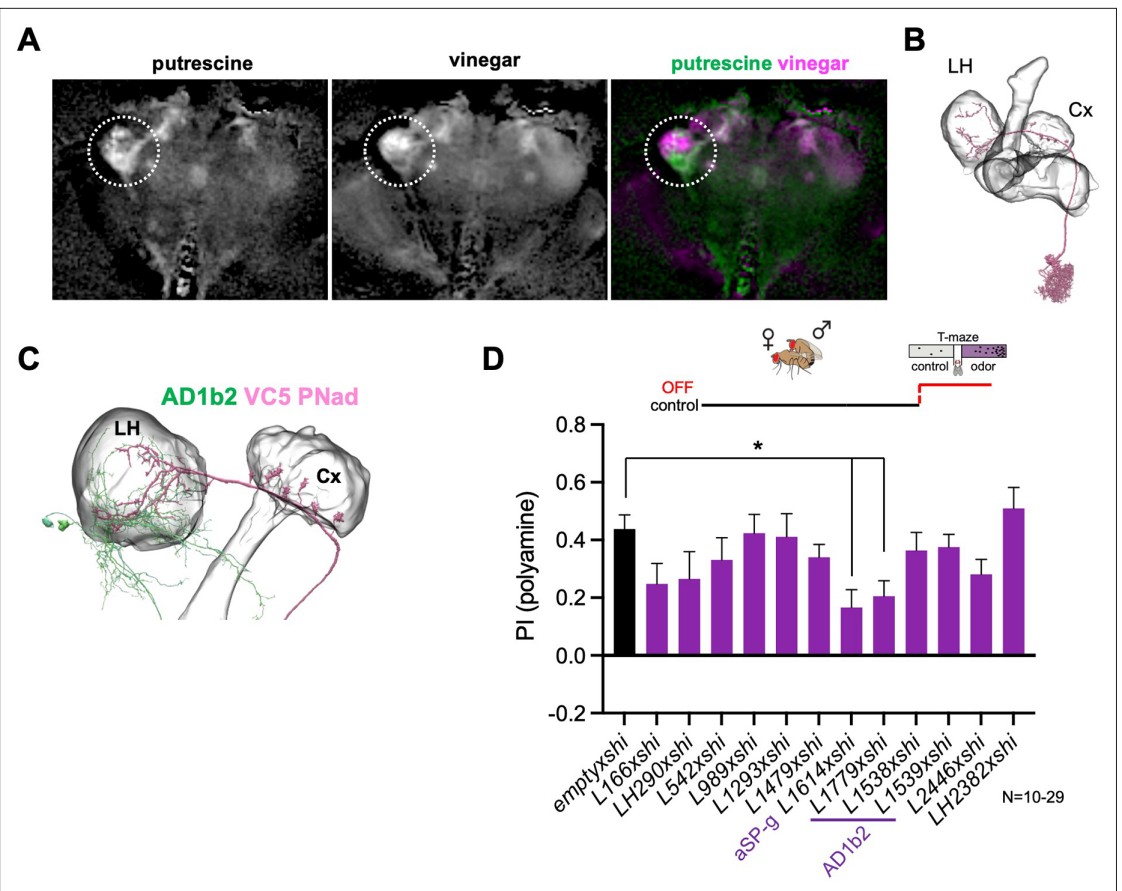

**Figure 5.** Lateral horn pathway contributes to the expression of polyamine attraction. (**A**) In vivo whole brain lightfield calcium imaging upon stimulation with putrescine or vinegar (*nsyb-Gal4; UAS-GCaMP6m*). Putrescine and vinegar activate different areas in the lateral horn (LH, dotted line). (**B**) Electron microscopy (EM)-based reconstruction showing projection neurons (PN) innervating the VC5 glomerulus. (**C**) VC5 PNad (pink) innervates the LH and the mushroom body (MB). (**D**) T-maze behavioral screen at restrictive temperature at test (30 °C) of candidate LHONs in mated females reveals that two LHON types are required for full attraction to putrescine. 'empty x shi' represents the control group and stands for *pBDPGAL4;UAS-shi^ts1^*, the Gal4 vector used to generate all MB-Gal4 lines without an expression driving enhancer. One-way Anova, Bonferroni corrected; p-values: *:<0.05.

The online version of this article includes the following figure supplement(s) for figure 5:

**Figure supplement 1.** Projection neuron and lateral horn neuron connectivity.

previously observed (*Strutz et al., 2014*). This suggested that putative polyamine-responding LHONs strongly innervate the ventral portion of the LH. We again used EM connectomics to search for LHNs receiving input from VC5 PNs in the EM connectome (*Figure 5B–D*, *Figure 5—figure supplement 1C*; *Li et al., 2020a*; *Scheffer et al., 2020*). According to neuPrint, three different types of PNs receive direct input from the VC5 glomerulus, adPN, lvPN, and l2PN. Only one of these types, adPN, innervates both the LH and the MB calyx. The primary LHON type receiving adPN input is AD1b2 (*Figure 5—figure supplement 1C*). Most AD1b2 dendrites are indeed located in the ventral part of the LH (*Figure 5C*). To test the function of putative LHNs more directly in polyamine attraction of mated females, we selected several LHN split-Gal4 lines that labeled AD1b2, but also LHNs that were not directly connected to VC5 PNs (*Dolan et al., 2019*). Using these LHN split-Gal4 lines, we temporally blocked synaptic output with Shi^ts1^ during test (*Figure 5D*). Only two lines showed a significant reduction (*Figure 5D*). One of these lines, L1779, strongly labeled several neurons of the AD1b2 type suggesting that the observed synaptic connections are indeed relevant for the attraction of putrescine. Two additional lines that also expressed in AD1b2 neurons, by contrast, did not reduce putrescine attraction significantly (*Figure 5D*). These lines, however, labeled fewer neurons than L1779 suggesting a certain redundancy between the AD1b2 type neurons (*Figure 5—figure supplement*

*1E-G*; L1538/1539: 9.8±1.5 *vs.* L1779: 19.3±2.6, p<0.001). Of note, AD1b2 neurons provide direct and indirect synaptic input to the β'1 compartment of the MB (*Figure 5—figure supplement 1H*).

Interestingly, expression of Shi$^{ts1}$ under the control of line L1614, which labels the aSP-g neuron, also significantly reduced the attraction to putrescine (*Figure 5D*). The aSP-g neuron receives input from the DA1 glomerulus (*Figure 5—figure supplement 1D*), which in turn receives input by the male pheromone cVA-detecting OSNs. This result is in line with the reduction in polyamine attraction observed by inactivation of ORCO- and OR67d-expressing OSNs (see *Figure 1*). Moreover, similar to AD1b2, aSP-g neurons provide indirect input to DANs innervating the β'1 MB-compartment (*Figure 5—figure supplement 1H-J*).

Altogether, these data indicate that multiple LHONs are involved in the mated female's attraction to polyamines, including AD1b2, which receives strong synaptic input by one type of VC5 PN, and aSP-g neurons, which are activated by cVA as well as some other odors (*Kohl et al., 2013*).

## Calcium imaging of MBON activity before and after mating

Our behavioral data suggested that β'1 MBONs play an important role in inducing mating state-dependent odor attraction. We used in vivo 2-photon calcium imaging to ask whether MBONs are modulated by the mating experience. Importantly, given a previous report that β'1 MBONs, in contrast to β'2 and other MBONs, respond only to few odors, we tested their response to cVA and putrescine (*Hige et al., 2015*). To this end, we recorded GCaMP-reported calcium changes in response to odor stimulation (*MB57B-Gal4;UAS-GCaMP6f*) and found that this MBON indeed responded to the odor of cVA, but not to the odor of putrescine, in line with a more specific or selective role of this MBON (*Figure 6A–E*). We next compared the response in virgins to the response in mated females 3–5 days after mating as done for the behavior. We did not find a significant difference between the response to cVA in virgins as compared to mated females (*Figure 6E*). In addition, neither MBON of virgins nor of mated females responded significantly higher to putrescine than to water (*Figure 6E*). These data showed that β'1 MBONs respond to cVA and that mating does not lead to a long-lasting modulation of this response. It is possible, however, that mating leads to a transient, short-lasting plasticity in odor response that we missed during our recordings.

Associative learning, that is, association of a shock with an odor, modulates the odor response of β'2 MBONs (*Owald et al., 2015*). Given the requirement of β'2-MBONs for the expression of polyamine preference, we investigated the response of the β'2 MBONs before and after mating. In contrast to β'1 MBONs, these MBONs responded equally strong to putrescine and cVA (*Figure 6F–H*). Different from our predictions that mating would modulate the polyamine response of the MBON, we again found no evidence for a long-lasting mating-induced modulation (*Figure 6H*). As for MBON-β'1, however, a mating-induced change in odor responsiveness, not lasting for several days and thus no longer detectable at the time of our recordings, is conceivable.

We next tested the effect of exogenous activation of PAM β'1,γ3/4 DANs, which is sufficient to induce a high polyamine attraction in virgins, on β'1 and β'2 MBONs (*Figure 6—figure supplement 1A*). We optogenetically activated these DANs with pulsing light for 24 hr using CsChrimson and recorded calcium responses in the β'1 and β'2 MBONs to putrescine odor thereafter (*MB188B-Gal4;UAS-CsChrimson; VT1211-lexA;lexAop-GCaMP7f*). Imaging following this exogenous activation again did not reveal a difference in polyamine responses between the test group and the controls (*Figure 6—figure supplement 1A*). Acute optogenetic activation of PAM β'1,γ3/4 at the time of imaging, however, led to an activation of β'1 and β'2 MBONs (*Figure 6—figure supplement 1B,C*). Although additional experiments would be needed as further confirmation, these data are consistent with the presence of an excitatory synapse between these DANs and the MBONs in addition to a neuromodulatory connection.

Based on the behavioral data, we predicted and found that MBON β'1 would respond to cVA. We also predicted that mating or the exogenous activation of β'1,γ3/4-PAMs would lead to a lasting modulation of the MBON odor response. However, we did not observe a lasting change in the response of β'1 or β'2 MBONs to putrescine after mating or exogenous PAM activation. We considered two main reasons for this result: First, it is possible that modulation of the β'1-compartment is only short-lasting (e.g. minutes to hours but not days) and therefore, any change is no longer observable at the time of imaging. In addition, the fact that β'1-MBONs bidirectionally modulate the behavior of virgin and mated females suggests that their role and thus the way they are modulated by DANs might be more

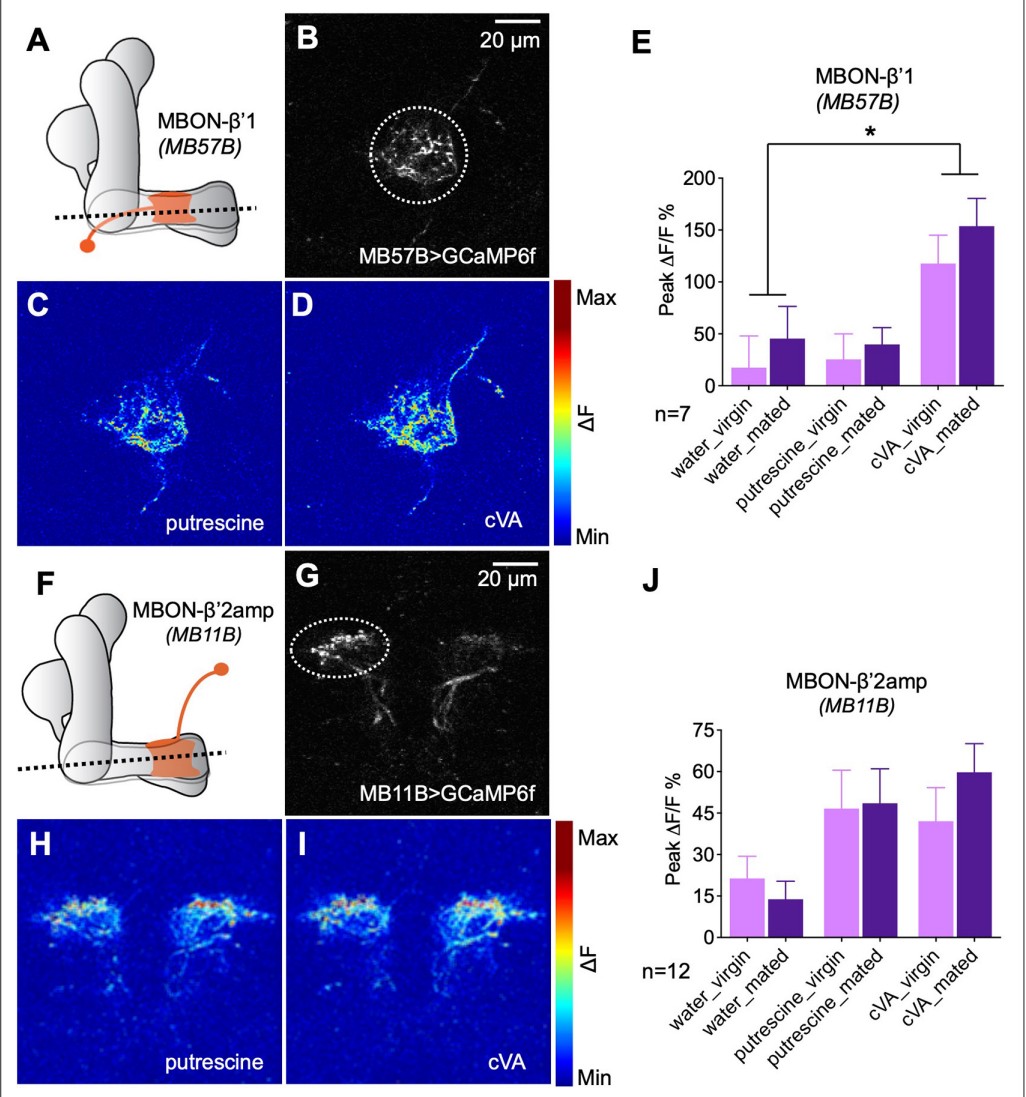

**Figure 6.** Odor responses of MBONs before and after mating. (**A**) Scheme depicting imaging plane through MBON-β'1. (**B**) GCaMP6f was expressed under the control of MB57B-Gal4 in β'1 MBONs. Dashed circle indicates the region of interest (ROI) used for quantification of the signal. (**C**) Representative image showing response of MBON-β'1 to putrescine. (**D**) Representative image showing the response of MBON-β'1 to cVA (11-cis-Vaccenyl Acetate). (**E**) Peak ΔF/F GCaMP signals in response to water, putrescine or cVA. Responses of virgins and mated females are shown. (**F**) Scheme depicting imaging plane through MBON-β'2. (**G**) GCaMP6f was expressed under the control of MB11B-Gal4 in β'2 MBONs. Dashed circle indicates the region of interest (ROI) used for quantification of the signal. (**H**) Representative image showing response of MBON-β'2 to putrescine. (**I**) Representative image showing the response of MBON-β'2 to cVA (cis-Vaccenyl Acetate). (**J**) Peak ΔF/F GCaMP signals in response to water, putrescine or cVA. Responses of virgins and mated females are shown. ANOVA, p-values: n.s.:>0.05, *:<0.05, **:<0.01, ***:<0.001.

The online version of this article includes the following figure supplement(s) for figure 6:

**Figure supplement 1.** Optogenetic manipulation paired with calcium imaging of mushroom body output neuron.

complex and not detectable through calcium imaging. Second, it is possible that neurons downstream of the MB involved in the expression of the behavior are lastingly modulated.

Thus, we next analyzed the response of the AD1b2 LH neuron in virgins as compared to mated females (*Figure 7*). Given that several neurons are labeled in the AD1b2 Gal4 line with slightly different projections, we imaged the odor response at the level of the cluster of cell bodies (*Figure 7*, upper left panel). In virgins, in vivo Ca²⁺-imaging indicated that AD1b2 neurons responded at the same

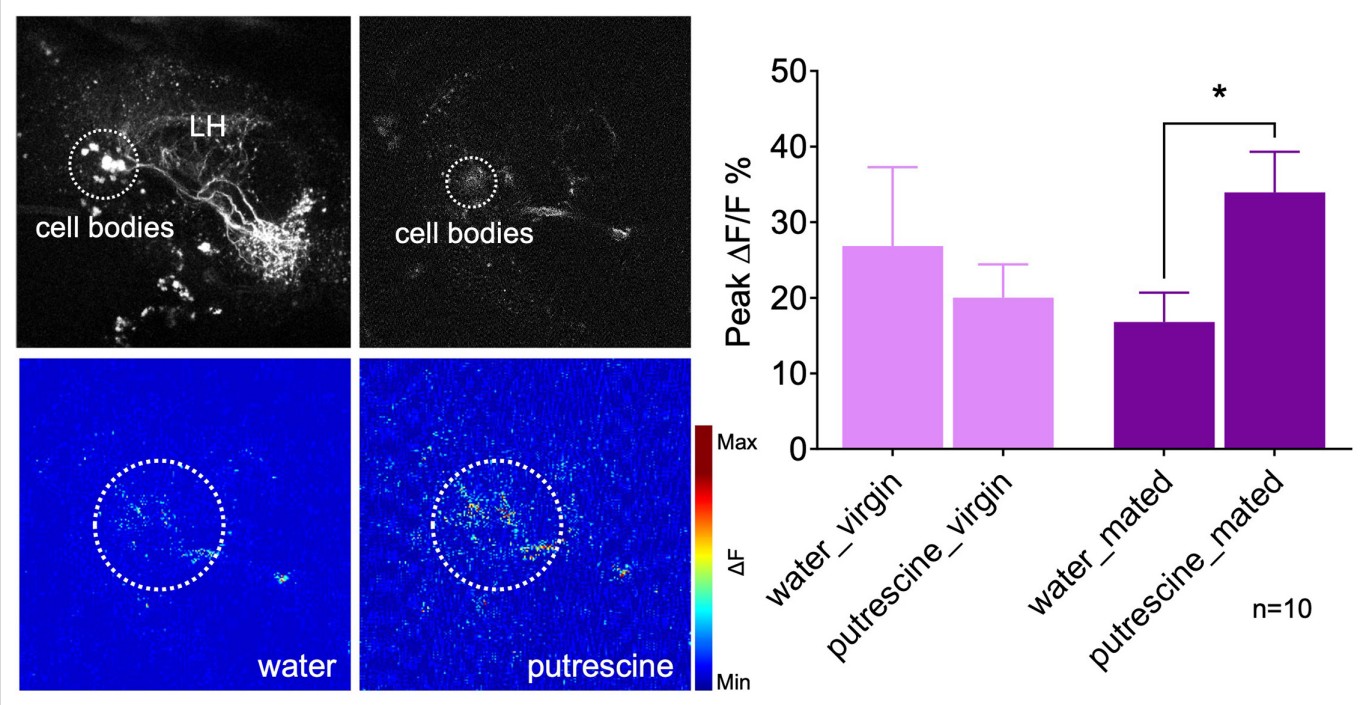

**Figure 7.** Mating modulates the putrescine response of lateral horn neuron AD1b2. Upper left panels: High-resolution two-phtoton image of expression of Gal4-line LH1779 (*LH1779-Gal4; UAS-GCaMP7s*) in the region of the lateral horn (LH). Lower left panels: Representative images showing response of the neurons labeled by LH1779 to water control and to putrescine. Right panels: Peak ΔF/F% GCaMP signals in response to water or putrescine in virgins or mated females. Dashed circles indicate the region of interest (ROI) used for quantification of the signal. ANOVA, p-values: n.s.:>0.05, *:<0.05.

The online version of this article includes the following figure supplement(s) for figure 7:

**Figure supplement 1.** Mushroom body output to lateral horn is involved in polyamine choice behavior.

level to the water control as to putrescine (*Figure 7*). In mated females, by contrast, AD1b2 neurons responded significantly stronger to putrescine than to the water control (*Figure 7*). These data are in line with the behavioral data where only mated females show a preference for putrescine over control, while virgins appear to be indifferent and do not prefer either odor. Moreover, these findings resemble previous work indicating that learning modulates the odor response of specific LH neurons (*Dolan et al., 2018*).

These data provide evidence that AD1b2 LHONs are lastingly modulated by mating. This modulation and our finding that they are involved in the expression of putrescine attraction in mated females indicates that mating-induced modulation of the LH is important for female reproductive behavior. How the MB and LH communicate is currently not clear. Inhibition of MBON-α2sc, which connects the MB to specific neurons in the LH, reduces polyamine attraction in mated females (*Figure 7—figure supplement 1*) consistent with the hypothesis that a mating-induced modulation of MB DANs could be relayed to the LH.

## Discussion

The transition from a sexually immature to a mature, reproducing parent is a major step in the life of most animals. With this step come new needs and demands, reflected in a significant change in an animal's behavior and preferences. While many aspects of reproductive behavior are innate, sexual and parental behavior is highly flexible and changes through physiology and with experience (*Griffith and Ejima, 2009*; *Koch and Ehret, 1989*; *Marlin et al., 2015*). In addition, sexually mature animals spend a major proportion of their lives with generating or caring for their offspring. In line with this, some of the behavioral and neuronal changes persist for a significant fraction of an animal's lifespan (*Hoekzema et al., 2017*; *Insel et al., 1995*; *Reisenman et al., 2009*; *Zhao et al., 2017*).

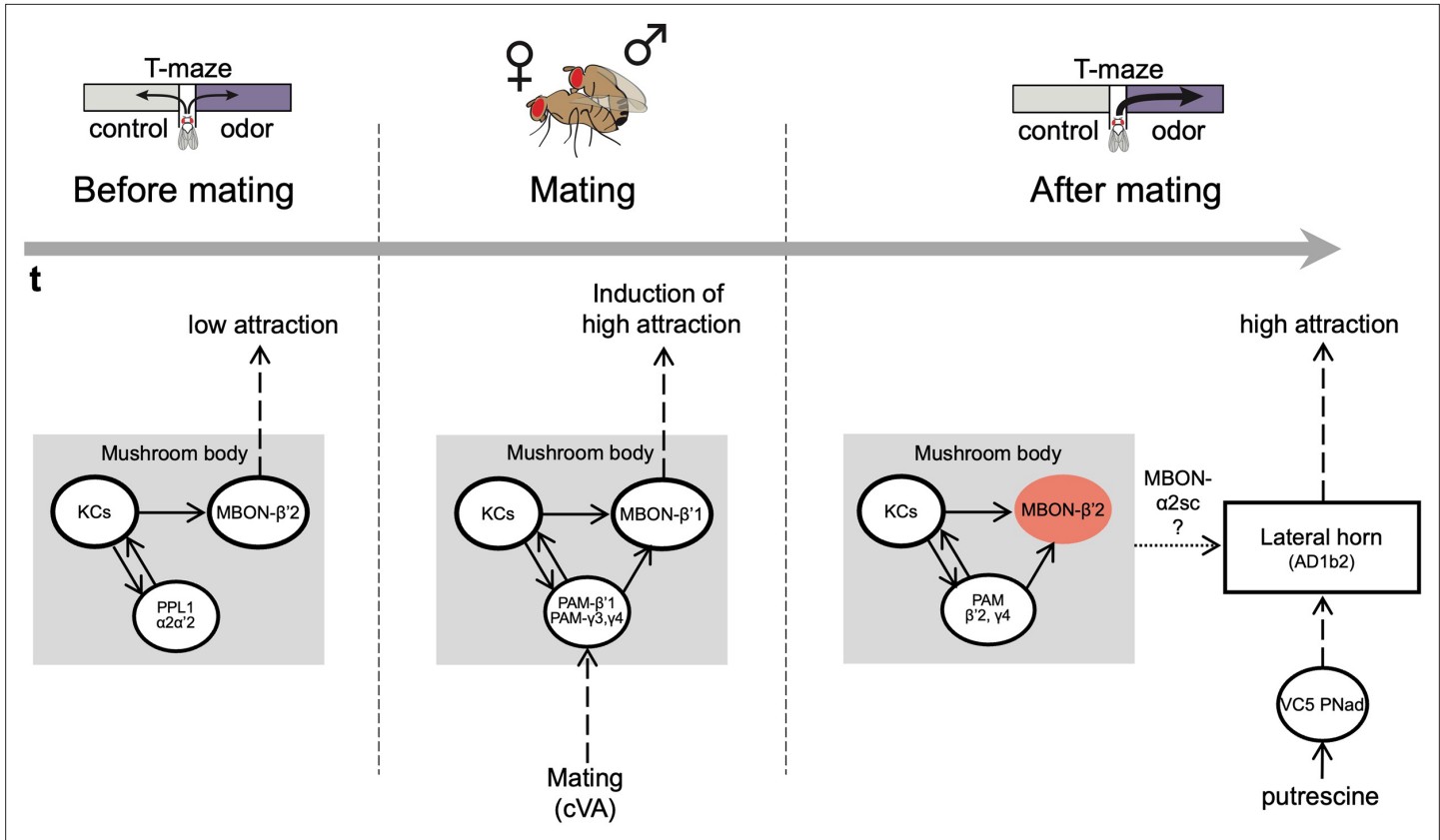

**Figure 8.** The mushroom body regulates odor preference behavior of females before and after mating. Working model showing that in virgin females (before mating), the activity of PPL1 dopaminergic neurons and MBONβ'2 suppress the expression of polyamine odor preference. During mating, β'1,γ3/4 PAM dopaminergic neurons and β'1-MBONs convey the experience of mating, possibly through the detection of the pheromone cVA, to the mushroom body. This mechanism induces a long-lasting change in polyamine preference in mating females. Moreover, mating changes the polyamine odor response of the AD1b2 LHONs. After mating, mated females express their higher attraction to polyamine odor through a circuit involving the activity of PAM-β'2,γ4 dopaminergic neurons and the suppression of MBON-β'2 output. In addition, output of the lateral horn through at least two different types of output neurons (AD1b2 and aSP-g) is required for the expression of high polyamine preference. It is conceivable that the lateral horn and the mushroom body circuits are connected through specific output neurons such as MBON-α2sc.

Here, we have identified a mechanism how a change in reproductive state translates into behavioral output. We propose a model wherein mating experience, through the sensation of mating-related odors such as cVA, is relayed to the MB via specific DANs innervating the β'1 compartment. Thereby, mating-related stimuli modulate or activate the output of β'1, and lastingly change the polyamine response of LHON AD1b2. This in turn induces a higher polyamine preference upon mating (*Figure 8*). The expression of this higher preference relies on β'2 MBONs and AD1b2 LHONs, analogous to the situation found upon associative learning (*Dolan et al., 2018*). Thus, changes induced by the experience of mating are induced and expressed through a dopamine-gated learning circuit. Given that neither MBONs nor LHONs are selective for any one odor, we propose that this mechanism could also play a role in the mating state-dependent adaptation of other choice behaviors.

## Triggers for reproductive state-dependent changes in the female brain

We have previously shown that SP appears to play a rather redundant role in the increase in polyamine attraction (*Hussain et al., 2016a*). Instead, SPR in OSNs together with its conserved ligand MIP depress OSN synaptic output upon mating and thereby reduce OSN sensitivity to polyamines. As a consequence, SPR mutant females do not show the increased preference for polyamines after mating (*Hussain et al., 2016a*). Mating also reduces the response of the DA1 glomerulus to cVA (*Lebreton et al., 2014*). How this translates into a higher response of β'1 DANs to cVA is unclear and probably lies within the complex, recurrent circuit structure of the MB. Nevertheless, the combined evidence indicates a mating-induced modulation of early as well as higher olfactory processing in MB and LH.

Our data reported here is consistent with the interpretation that olfactory signals related to mating are involved in conveying mating experience to higher olfactory processing as transient blocking of synaptic output of ORCO-dependent OSNs (i.e. OR67d) during mating reduced the mated females' attraction at the time of test, suggesting that OR-dependent pheromone detection contributes to the change in behavior from virgin to mated female. Of note, males transfer some of their pheromone onto the female during copulation (*Keleman et al., 2012*), and they also mark feeding and oviposition sites for females by depositing pheromone (*Lin et al., 2015*), suggesting that pheromones are present in the female's environment just before, during and sometime after mating (*Ejima et al., 2007*). By contrast, the presence of polyamines during mating was not required as temporal inhibition of polyamine OSN synaptic output during mating did not affect the mated female's high putrescine attraction during test. In line with a putative role of odors such as pheromones, in vivo calcium imaging data shows that the pheromone cVA activates DANs of the β'1 compartment; this activation is modulated by state (*Siju et al., 2020*). We speculate, that the mechanisms we describe here might represent a female version of male courtship learning (*Ejima et al., 2007*; *Keleman et al., 2012*). Consistent with our observations, specific DANs can mimic courtship experience in naive males in the absence of an actual rejection by an already mated female (*Keleman et al., 2012*). Males likely associate cVA, which was transferred during previous copulation by another male onto the female, with the experience of rejection by an already mated female (*Ejima et al., 2007*). In our scenario, the pheromones transferred to the female during copulation would be 'sensed' by the β'1 DANs which in turn induce a shift in how the MB regulates the expression of odor preference through output neurons of the LH (*Figure 8*).

Naturally, mating itself stimulates also mechanosensory neurons in the female's reproductive tract and thereby appears to contribute to a switch in her behavior (*Gou et al., 2014*; *Shao et al., 2019*). Among the most important is SPR-signaling (*Yapici et al., 2008*). SPR mutant females, as mentioned above, do not undergo the change in polyamine preference (*Hussain et al., 2016a*). Other hormonal or peptidergic signals, such as steroids or juvenile hormone (JH) could play a role (*Moshitzky et al., 1996*). For instance, JH drives network maturation of the MB in young flies and thereby promotes their ability to form memories (*Leinwand and Scott, 2021*). Recent elegant work showed that mating triggers neuropeptide (i.e. Bursicon) release from enteroendocrine cells, which in turn modulate female appetite (*Hadjieconomou et al., 2020*). Thus, additional factors are likely involved in conveying mating experience to the female's higher brain centers. We propose that these factors also explain the fact that inhibition of PAM-β'1,γ3/4 during mating does not prevent the switch in the mated female's preference for polyamines. Thus, PAM-β'1,γ3/4 is sufficient to induce a behavior in a virgin female that resembles a specific aspect of the behavior of a mated female, but it not strictly necessary. Given the importance of reproductive behavior for any species, it might not be surprising that multiple parallel systems ensure the transition from a virgin to a mated female.

## Mushroom body and lateral horn output promote state-dependent female preference behavior

The finding that KC synaptic output is required during mating for a lasting change in female choice behavior and not only for its expression is particularly interesting. Previous results using associative learning paradigms suggested that KCs are not essential during learning but only during test (*Dubnau et al., 2001*; *McGuire et al., 2001*; *Schwaerzel et al., 2002*).

In line with our findings, a very recent study suggests that KC output is necessary during training to establish appetitive odor memories (*Pribbenow et al., 2022*).

Like KCs, inhibition of MBON-β'1 output at the time of mating prevented the switch to mated female behavior. Similarly, activation of MBON-β'1 output instead of mating also increased the virgin's polyamine attraction to mated female levels. We thus propose that mating-induced modulation of the relative activity and output of the β'1 compartment of the MB changes the female's interest in the beneficial compound polyamine. Notably, β'1 MBONs have been implicated previously in courtship conditioning in *Drosophila* males (*Montague and Baker, 2016*). Our data confirms that these MBONs are quite selective in their odor response (*Hige et al., 2015*). They don't respond to putrescine but to cVA. At this point, we can only speculate about what β'1 MB output does to reproduction related behavior. EM connectomics and trans-TANGO data suggest that β'2 DANs are directly and β'2 MBONs indirectly connected to MBON-β'1. In line with this, we find that blocking β'2 MBON output

increases the virgin's and exogenous activation reduces the mated females attraction to polyamine at the time of test.

If the mating-induced long-lasting changes in polyamine odor preference are due to a learning mechanism, why do we only see lasting mating-induced changes in the odor response in β'1-DANs but not in β'1 or β'2 MBONs? One possible explanation could be that the changes are subtle and would require the analysis of a significantly higher number of animals. Alternatively, they could be short-lived (e.g. minutes or hours) and observable only at a very specific time point in these MBONs after mating that we were not fortunate enough to hit in our imaging experiments. It is also conceivable that, as the behavioral data indicates, the role of the β'1 compartment is complex and bidirectional, possibly with subtypes of β'1-innervating DANs that connect to different downstream neurons and receive different inputs (see also *Figure 3—figure supplement 1D*, *Figure 5—figure supplement 1H-J*). Mating could affect these subtypes differently such that a difference is difficult to identify at the current resolution of split-Gal4 lines.

Lastly, lasting modulation could manifest in changes downstream of the MB circuit. Like previous work indicating that the expression of associative learning requires the activity of specific LHONs (*Dolan et al., 2018*), we also find that AD1b2 LHONs are involved in the expression of putrescine attraction in mated females. As our data suggests, these neurons appear to undergo longer lasting changes in their response to putrescine upon mating (see *Figure 7*). While AD1b2 neurons in virgins do not respond differently to the water control as compared to putrescine, mated females do. Given that these neurons are involved in the expression of polyamine attraction, we favor a model where state-dependent changes in the MB network are relayed to the LH to modulate odor attraction. How a mating-induced modulation of the β'1 compartment is relayed to the LH remains to be further investigated. It is possible that another MBON, MBON-α2sc, which connects from the MB to specific neurons in the LH, is involved (see *Figure 7—figure supplement 1*; *Dolan et al., 2018*).

## Does state change or a discrete signal induce neuronal changes to facilitate reproductive behaviors?

A state change, such as before and after mating, could conceivably represent a window of opportunity for changes in relevant neural networks (*Griffith and Ejima, 2009*) or even globally in many or all brain regions. Our in vivo whole brain imaging data suggests that, at least in female flies, mating does not induce a strong global change in brain state. Rather, mating appears to induce discrete changes in a dopamine-gated learning circuit. In line with this interpretation, mating and SPR signaling were recently shown to enhance MB-dependent long-term memory in *Drosophila* females (*Scheunemann et al., 2019*). Similar scenarios as through mating might arise at times of hunger (*Krashes et al., 2009*; *Root et al., 2011*). In other words, food consumed when hungry not only tastes and smells better, it also leaves a longer lasting impression. Mechanistically, such a window of enhanced plasticity might arise through an increase in cAMP in neurons of the MB network at critical phases of life (*Louis et al., 2018*). Our data provides some evidence that a discrete signal (e.g. pheromone, the act of courting and mating) rather than the physiological state change itself induces the observed increase in olfactory preference. Females that cannot produce eggs, were mated to sterile males, or had a small group of neurons activated instead of mating, they all displayed the increased attraction to polyamines. Given the variety of behavioral adaptations and their ethological importance, it is conceivable that a combination of discrete signals and systemic physiological changes, possibly dependent on the type of behavioral change, work hand-in-hand or in parallel to promote the right behavior at the right time.

## Ideas and speculation

While not all odor responses or odor-induced behaviors are subject to mating-induced changes, it is conceivable that other mating-induced adaptations in sensory including odor perception use the same or similar mechanism as described here on the example of polyamines. Importantly, odor responses in higher brain centers show relatively little odor selectivity for a specific odorant (*Frechter et al., 2019*; *Hige et al., 2015*). MBONs and LH neurons usually respond to subgroups of odorants possibly depending on their innate or experience-dependent valence. Therefore, it is unlikely that the described mechanism, including the shown manipulations of specific neurons (e.g. manipulations during odor preference test), work exclusively in the context of mating state-dependent behavior to

polyamines. We speculate that the described mechanism could be a more general one allowing the responses to other odors or possibly even other sensory modalities to adapt upon mating.

Beyond insects, such a scenario could arise during different periods of important changes in the body and its environmental demands (*Carlson et al., 2018*). An example could be mother-child-bonding in mammals where mother and newborn are particularly sensitive to memorizing the sensory characteristics of each other for a certain period after birth (*Cernoch and Porter, 1985*; *Galbally et al., 2011*; *Sullivan, 2003*). For instance, the neuropeptide oxytocin, which is released during birth and lactation, induces lasting changes in the mother's cortex, transforming weaker responses to pup calls in the virgin to strong responses in mothers (*Marlin et al., 2015*). The orchestration of internal state-dependent sensory tuning with plasticity of neurons in higher brain centers could, thus, ensure that animals remember the most relevant information at key turning points in their lives.

# Materials and methods

**Key resources table**

| Reagent type (species) or resource | Designation | Source or reference | Identifiers | Additional information |
|---|---|---|---|---|
| Antibody | Anti-Mouse Alexa488, goat polyclonal | Molecular Probes | AB_ 150113 | (1:200) |
| Antibody | Anti-Mouse Alexa633, goat polyclonal | Molecular Probes | AB_141431 | (1:200) |
| Antibody | Anti-Rabbit Alexa568, goat polyclonal | Molecular Probes | AB_141416 | (1:200) |
| Antibody | Anti-Rabbit Alexa633, goat polyclonal | Molecular Probes | AB_2535731 | (1:200) |
| Antibody | Anti-Rat Alexa568, goat polyclonal | Molecular Probes | AB_141874 | (1:200) |
| Antibody | Anti-ChAT, mouse monoclonal | *Yasuyama et al., 1995* | N/A | (1:100) |
| Antibody | Anti-OA, mouse monoclonal | Jena Bioscience | AB_2315000 | (1:200) |
| Antibody | Anti-dsRed, rabbit polyclonal | Clontech | AB_10013483 | (1:400) |
| Antibody | Anti-GFP [3H9], rat monoclonal | Chromotek | AB_10773374 | (1:200) |
| Antibody | Anti-Ncadherin, rat monoclonal | DSHB | AB_528121 | (1:200) |
| other | All original data is available online. | Mendeley Data | http://dx.doi.org/10.17632/5rz28jr8gc.1 | Source data |
| genetic reagent (*D. melanogaster*) | D.mel/Canton-S | Bloomington DSC | Flybase: FBst0064349 | |
| genetic reagent (*D. melanogaster*) | D.mel/GMR95A10-LexA | Bloomington DSC | FlyBase: FBst0061633 | |
| genetic reagent (*D. melanogaster*) | D.mel/LexAop2-mCD8-GFP | Bloomington DSC | FlyBase: FBst0056182 | |
| genetic reagent (*D. melanogaster*) | D.mel/LexAop2-GCaMP7f | Bloomington DSC | FlyBase: FBti0202377 | |
| genetic reagent (*D. melanogaster*) | D.mel/MB10B | Janelia RC | Flybase: FBst0068293 | *Aso et al., 2014a* |
| genetic reagent (*D. melanogaster*) | D.mel/MB11B | Janelia RC | Flybase: FBst0068294 | *Aso et al., 2014a* |
| genetic reagent (*D. melanogaster*) | D.mel/MB438B | Janelia RC | Flybase: FBst0068326 | *Aso et al., 2014a* |
| genetic reagent (*D. melanogaster*) | D.mel/MB57B | Janelia RC | Flybase: FBst0068277 | *Aso et al., 2014a* |

*Continued on next page*

*Continued*

| Reagent type (species) or resource | Designation | Source or reference | Identifiers | Additional information |
|---|---|---|---|---|
| genetic reagent (*D. melanogaster*) | D.mel/MB60B | Janelia RC | Flybase: FBst0068279 | *Aso et al., 2014a* |
| genetic reagent (*D. melanogaster*) | D.mel/MB58B | Janelia RC | FlyBase: FBst0068278 | *Aso et al., 2014a* |
| genetic reagent (*D. melanogaster*) | D.mel/MB80C | Janelia RC | FlyBase: FBst0068285 | *Aso et al., 2014a* |
| genetic reagent (*D. melanogaster*) | D.mel/MB188B | Janelia RC | gift of Yoshinori Aso | *Aso et al., 2014a* |
| genetic reagent (*D. melanogaster*) | D.mel/MB42B | Janelia RC | FlyBase: FBst0068303 | *Aso et al., 2014a* |
| genetic reagent (*D. melanogaster*) | D.mel/MB109B | Janelia RC | gift of Yoshinori Aso | *Aso et al., 2014a* |
| genetic reagent (*D. melanogaster*) | D.mel/MB316B | Janelia RC | gift of Yoshinori Aso | *Aso et al., 2014a* |
| genetic reagent (*D. melanogaster*) | D.mel/L166 | Janelia RC | gift of Greg Jefferis | *Dolan et al., 2019* |
| genetic reagent (*D. melanogaster*) | D.mel/L290 | Janelia RC | gift of Greg Jefferis | *Dolan et al., 2019* |
| genetic reagent (*D. melanogaster*) | D.mel/L542 | Janelia RC | gift of Greg Jefferis | *Dolan et al., 2019* |
| genetic reagent (*D. melanogaster*) | D.mel/L989 | Janelia RC | gift of Greg Jefferis | *Dolan et al., 2019* |
| genetic reagent (*D. melanogaster*) | D.mel/L1293 | Janelia RC | gift of Greg Jefferis | *Dolan et al., 2019* |
| genetic reagent (*D. melanogaster*) | D.mel/L1479 | Janelia RC | gift of Greg Jefferis | *Dolan et al., 2019* |
| genetic reagent (*D. melanogaster*) | D.mel/L1614 | Janelia RC | gift of Greg Jefferis | *Dolan et al., 2019* |
| genetic reagent (*D. melanogaster*) | D.mel/L1538 | Janelia RC | gift of Greg Jefferis | *Dolan et al., 2019* |
| genetic reagent (*D. melanogaster*) | D.mel/L1539 | Janelia RC | gift of Greg Jefferis | *Dolan et al., 2019* |
| genetic reagent (*D. melanogaster*) | D.mel/L1779 | Janelia RC | gift of Greg Jefferis | *Dolan et al., 2019* |
| genetic reagent (*D. melanogaster*) | D.mel/L2446 | Janelia RC | gift of Greg Jefferis | *Dolan et al., 2019* |
| genetic reagent (*D. melanogaster*) | D.mel/L2382 | Janelia RC | gift of Greg Jefferis | *Dolan et al., 2019* |
| genetic reagent (*D. melanogaster*) | D.mel/pBDP-Gal4U (,empty-Gal4') | Bloomington DSC | Flybase: FBst0068384 | |
| genetic reagent (*D. melanogaster*) | D.mel/UAS-CsChrimson | Bloomington DSC | FlyBase: FBst0055134 | |
| genetic reagent (*D. melanogaster*) | D.mel/UAS-DenMark | Bloomington DSC | FlyBase: FBst0033062 | |
| genetic reagent (*D. melanogaster*) | D.mel/UAS-GCaMP6f | Bloomington DSC | FlyBase: FBst0042747 | |
| genetic reagent (*D. melanogaster*) | D.mel/UAS-GCaMP7s | Bloomington DSC | FlyBase: FBst0079032 | |

*Continued on next page*

*Continued*

| Reagent type (species) or resource | Designation | Source or reference | Identifiers | Additional information |
|---|---|---|---|---|
| genetic reagent (*D. melanogaster*) | D.mel/UAS-mCD8-GFP | Bloomington DSC | FlyBase: FBst0030001 | |
| genetic reagent (*D. melanogaster*) | D.mel/UAS-Shibire[ts1] | Bloomington DSC | FlyBase: FBst0044222 | |
| genetic reagent (*D. melanogaster*) | D.mel/UAS-syt-GFP | Bloomington DSC | FlyBase: FBst0006926 | |
| genetic reagent (*D. melanogaster*) | D.mel/w1118 | Bloomington DSC | Flybase: FBst0003605 | |
| genetic reagent (*D. melanogaster*) | D.mel/prd-gal4 | | gift of Anne von Philipsborn | |
| genetic reagent (*D. melanogaster*) | D.mel/UAS-Bip RNAi | | gift of Anne von Philipsborn | |
| genetic reagent (*D. melanogaster*) | D.mel/tud[1] bw[1] sp[1]/CyO, l(2)DTS513[1] | Bloomington DSC | stock # 1786 | |
| genetic reagent (*D. melanogaster*) | D.mel/ovoD | Bloomington DSC | stock # 38444 | |
| Software/ algorithm | Matplotlib 1.4.2 | *Hunter, 2007* | https://matplotlib.org/ | |
| Software/ algorithm | Numpy 1.8 | *Harris et al., 2020* | https://numpy.org/ | |
| Software/ algorithm | Prism 6 and 7 | GraphPad | https://www.graphpad.com/scientific-software/prism/ | |
| Software/ algorithm | Python 2.7 | *Van and Fred, 1995* | https://www.python.org/ | |
| Software/ algorithm | Pyvttbl 0.5.2.2 | *Galbally et al., 2011* | https://github.com/rogerlew/pyvttbl | |
| Software/ algorithm | FIJI | ImageJ | https://imagej.net/software/fiji/ | |
| Software/ algorithm | Scipy.stats 0.14 | *Jones et al., 2001* | https://scipy.org/ | |
| Software/ algorithm | Igor Pro 6.37 | Wave Metrics | https://www.wavemetrics.com/ | |
| Software/ algorithm | NeuroMatic 3.0 | *Rothman and Silver, 2018* | http://neuromatic.thinkrandom.com/ | |
| Software/ algorithm | LAS AF E6000 and LAS X | Leica Microsystems | https://www.leica-microsystems.com/ | |
| Software/ algorithm | FV10-ASW | Olympus | https://olympus-lifescience.com/ | |
| Software/algorithm | ScanImage | Vidrio Technologies | https://vidriotechnologies.com | |

## Fly husbandry

Flies were kept at 25 °C at 60% humidity with a day:night cycle of 12 hr each. Flies were raised on standard cornmeal medium. Mushroom body lines were received from the Janelia Fly-Light Split-GAL4 Driver Collection (*Aso et al., 2014a*). BiP, prd lines were a generous gift from Anne von Philipsborn, trans-TANGO lines were a gift from the Barnea Lab, LH lines were kindly provided by Greg Jefferis.

## Experiments with wildtype flies

After eclosion, female virgins were kept in a vial until testing one week later. A second group of female flies were kept with males for 24 hr. Then, the males were removed, and the females were kept separately in a fresh vial until final testing 3–5 days later. For virgins and mated females, vials were checked for larvae to ensure mating did not or did take place, respectively.

To test the effect of cVA (11-cis-Vaccenyl acetate) exposure on virgin females, 50 µl cVA (pure, Pherobank) or as a control ddH$_2$O were deposited onto a small filter paper disc and added to the food vial containing 1- to 2-day-old virgins. The discs were placed between two layers of mesh to prevent flies from touching or tasting the solution on the disc. After exposure for 24 hr, the flies were transferred into a clean vial and tested in a T-maze 3 days later.

## Shibire and dTrpA1 experiments

For shibire[ts1]- and dTrpA1 -silencing or -activation, respectively, flies were shifted to a temperature of 30 °C for time frames indicated in the results section.

For temperature shifts *at mating*, 1- to 2-day-old virgin females were put on 30 °C for 30 min before experienced CantonS males were added for 24 hr at 30 °C to allow mating during this time period. The males were removed after 24 hr. Females were tested 3–5 days later for putrescine preference behavior at 25 °C in a T-maze assay. The vials were checked for offspring to ensure mating had taken place. For virgin experiments, virgins were kept for 24 hr at 30 °C without males. This was followed 3–5 days later for putrescine preference behavior at 25 °C in a T-maze assay.

For temperature shifts *at testing*, 1-day-old females were kept with males for 24 hr at 25 °C. Then, the males were removed, and females were tested 3–5 days later for putrescine preference behavior. Virgins were not exposed to males. Flies were tested in a preheated T-maze chamber at 30 °C and 60% humidity.

## T-maze experiments

The two-choice population assay or T-maze was performed as previously described in *Lewis et al., 2015*. T-maze tubes were either prepared with 50 µl of 100 mM 1,4-Diaminobutane (Putrescine, Sigma), 50 µl ultrapure water or different concentrations of ammonia on a piece of Whatman filter paper and sealed with Parafilm until right before the experiment started. Note that the concentration of putrescine is higher than previously reported in Hussain et al. due to a change odor solution preparation as follows. To ensure precise odor concentrations, solid 1,4-Diaminobutane was heated mildly until the solid pieces became liquid. The precise amount was pipetted directly from the solution and diluted in water. Odor solutions were stored at 4 °C for a maximum of 14 days, because of the rapid degradation of putrescine molecules in water and air. Thus, it is important to note that the precise concentration of 1,4-Diaminobutane varied and was usually lower than the initial concentration of 100 mM. To ameliorate the problem of fast molecule degradation, odor solution efficacy was regularly tested on CantonS wildtype mated and virgin females and remade if the expected attraction of mated females and low attraction of virgin females was not observed. For the test, flies were tested in groups of ~30 in a non-aspirated T-maze and were allowed 1 min to respond to stimuli. Experimentation was carried out in climate-controlled boxes at 30 °C and 60% rH. A preference index (PI) was calculated by subtracting the number of flies on the control side by the number of flies on the stimulus side and normalizing by the total number of flies. The data is represented by bar graphs of the PIs plus standard error of the mean (SEM) and was analyzed using a Mann-Whitney-U test in R-software. This has been done in Excel's RealStats Resource Pack. Unless indicated otherwise, significance stars or n.s. concern the comparison between the two virgin groups or the two mated fly groups, respectively.

Given the importance of reproductive state, we used a single fly assay in order to be able to confirm the reproductive state of each animal tested. All virgin flies that laid fertilized eggs in the days after an experiment were excluded from the analysis. Similarly, all mated females that did not lay fertilized eggs were equally excluded. For single fly T-maze experiments, single mated or virgin females were tested in the same T-maze setup as described above. They were given 1 min to make a choice between water and putrescine. The data is represented as percentage of flies that made a certain choice in bar graphs and was analyzed using a Fisher's exact test in R-software. This has been done in Excel's RealStats Resource Pack. Unless indicated otherwise, significance stars or n.s. concern the comparison between the two virgin groups or the two mated fly groups, respectively.

All experiments were carried out with the experimenter unaware of the genotype or state of the sample. In addition, most experiments were repeated independently by another person in the laboratory.

## Courtship experiments

Male flies were kept together in a vial for 2 days until exposure to 1–2 days old virgin females. Always one male was put into a vial with one female. The process of courtship as described in previous studies (*Dickson, 2008*) was monitored until the male began licking the female and making first attempts for copulation. Immediately, the male was removed from the vial. The female was tested 3–5 days later in a T-maze for putrescine attraction or avoidance behavior at 25 °C.

## Immunohistochemistry

Trans-TANGO flies were kept for 1 week at 25 °C followed by another 2 weeks at 18 °C, as similarly described in the original paper (*Talay et al., 2017*). Flies were anesthetized on ice, put in a glass staining cup with 80% ethanol. After 30 s flies were put in another glass staining cup filled with 1 x phosphate buffered saline (PBS). Flies were dissected under the microscope and brains were stored in a PCR cap with 1:4 solution of 4% paraformaldehyde (PFA) and 0.1% phosphate buffered triton (PBT) until final fixation.

Brains were fixed in PFA and a drop of PBT for 60 min at room temperature. Brains were then washed with PBT three times for 20 min at room temperature. The PBT was removed and replaced by 3% normal goat serum (NGS) for 30 min at room temperature. The first antibody mix was incubated in 1:200 with α-GFP (mouse), α-RFP (rabbit), α-NCAD (rat) and 3% NGS for 24 hr at 4 °C in darkness. Brains were washed with PBT for 5 s and then three times for 20 min at 4 °C in darkness. The second antibody mix was incubated at 1:200 with α-mouse alexa488, α-rabbit Cy3, α-rat alexa633 and 3% NGS for 24 hr at 4 °C in darkness. Brains were washed with PBT for 5 s and then three times for 20 min at 4 °C in darkness. After washing with PBT for one last time for 1 hr at 4 °C in darkness, brains were mounted on a glass slide with VectaShield. Imaging was carried out at a Leica SP8 confocal microscope. Image Processing and analysis have been performed using Fiji software (*Schindelin et al., 2012*).

## Electron microscopy connectomics

All images, connectomes and related data were obtained by working with the database NeuPrint (https://neuprint.janelia.org). Connectomes and reconstructed images only depict subsets of neurons and not all neurons that are synaptically connected to each other. Of note, we are showing all synaptic connections between neurons discussed and/or analyzed in the study, but not all existing or possible connections in the included graphs. Moreover, we have opted to not threshold the number of synapses as it is frequently done in other EM connectomics studies (e.g. *Hulse et al., 2021*), where only connections as of 10 synapses are shown.

## In vivo two-photon calcium imaging

All imaging experiments were conducted with a two-photon microscope. Five- to 7-day-old female virgin or mated flies of appropriate genotypes were used for experiments. In vivo fly preparations were prepared according to a method described previously. Experiments in *Figure 6* were imaged using an Olympus FV1000 two-photon microscope system with a BX61WI microscope and a 40 x, 0.8 water immersion objective. GCaMP fluorescence was excited at 910 nm by a mode-locked Ti:Sapphire Mai Tai DeepSee laser. Time series images were acquired at 200x200 pixel resolution at a rate of 3 frames per second for 200 frames using the Olympus FV10-ASW imaging software. In order to minimize brain movement of in vivo preparations under the two-photon microscope, a drop of 1% low melting temperature agarose (NuSieveGTG, Lonza) in imaging buffer maintained at 37 °C was added to the exposed brain. Experiments described in *Figure 7* and *Figure 6—figure supplement 1* were imaged using a custom-built two-photon microscope ('Denk scope') with a 40 x, 0.8 water immersion objective. Fluorescence were excited at 910 nm by a mode-locked Ti:Sapphire Coherent Chameleon laser. Microscope control and acquisition of images were done by ScanImage software. Time series images were acquired at 236x236 pixel resolution at a rate of 2 frames per second for 120 frames. A custom-made odor delivery system with mass flow controllers was used for odor stimulation. The odor was delivered in a continuous airstream (1000 ml/min) through an 8 mm Teflon tube placed ~1 cm away from the fly. 1% cVA diluted in paraffin oil (Pherobank, The Netherlands) and 100 mM putrescine diluted in water were used for stimulation. Changes in fluorescence intensity were measured in manually drawn regions of interest (ROI) using the Olympus FV10-ASW software or Fiji ImageJ. Relative changes in fluorescence intensity were defined as $\Delta F/F = 100*(F_i - F_0)/F_0$ for the i frames after stimulation. Fluorescence background, $F_0$, is the average fluorescence of 5 or 10 frames before the stimulus. Pseudocolored images were generated using a custom-written MATLAB program and ImageJ.

## In vivo light field calcium imaging

Female flies (*nsyb-Gal4;UAS-Gcamp6s*) were collected at eclosion to obtain virgins or mated for 48 hr with males of the same genotype. After the males were removed from the mated group, the labels of

both groups were blinded to avoid bias. Successful mating or virginity was confirmed later by keeping the vials at 25 °C and checking for larvae a few days after the experiment. Flies were prepared for whole brain imaging at Day 4 after eclosion, as previously described (*Woller et al., 2021*). The light field microscope was set up according to *Aimon et al., 2019*. The system was based on a Thorlabs Cerna with a Leica HC FLUOTAR L 25 x/0.95 objective and MLA-S125-f12 microlens array (RPC photonics). The microlens array was placed on the image plane, while the camera imaged the microlens array through 50mmf/1.4 NIKKOR-S Nikon relay lenses. The light field images were recorded at 10 Hz with a scientific CMOS camera (Hamamatsu ORCA-Flash 4.0). For odor delivery, a Syntech Stimulus Controller (CSS-55) was connected to a 4 mm PVC tube that was placed under the microscope, 5 mm in front of the fly's head and delivered a constant air stream of 1000 ml/min. Flies were stimulated by redirecting 30% of main air flow for 1 s through a head-space glass vial filled with 1 ml of 100 mM putrescine by a manual trigger. Each fly was stimulated five times with 50 s interstimulus intervals. The imaging data was processed as described in *Siju et al., 2020*. Volumes were reconstructed using a python program developed by *Broxton et al., 2013* (*Broxton et al., 2013*) and available on github: https://github.com/sophie63/FlyLFM*Aimon, 2021*. The movement artifacts were removed by 3D registration using the 3dvolreg routine from AFNI. The following steps were performed with MATLAB (version R2019b, MathWorks Inc) unless stated otherwise. The voxel time series were transformed to DF/F by subtracting and normalizing by a moving average over 20 s and noise was reduced, using Kalman filtering. Functional regions were then extracted using principal component analysis and independent component analysis. These were then used as landmarks to align the recordings to anatomical templates from *Ito et al., 2014* in FIJI. Masks were created to extract and average DF/F time series for twelve major brain regions (neuropil supercategories). Linear regression of the response time series with the stimulus time series (convolved with the response profile of GCamp6s) was performed. Peak responses to the first four odor stimulations were calculated by subtracting the mean DF/F in a 5 s window before the stimulus from the maximum value in the 20 s window after the stimulus. Interstimulus phases were extracted as 30 s intervals (10 s after a stimulus and 10 s before the next stimulus). Statistical analysis was performed with GraphPad Prism (version 9.1.2, GraphPad Software LLC).

## Chemical analysis of fly food

### Chemicals

The following compounds were obtained commercially from the sources given in parentheses: formic acid, sodium hydroxide (Merck, Darmstadt, Germany), histamine, ethanolamine, 2-phenylethylamine, putrescine, β-alanine, tyramine, spermine, spermidine, trichloroacetic acid (Sigma-Aldrich, Taufkirchen, Germany), benzoyl chloride (VWR, Darmstadt, Germany), [2H4]-histamine, [2H4]-putrescine (CDN Isotopes, Quebec, Canada), [13C2]-ethanolamine, [13C3,15N]-β-alanine (Sigma-Aldrich, St. Louis, MO, USA), [2H4]-phenylethylamine (Dr. Ehrenstorfer, Augsburg, Germany). The purity of all amines was checked by LC-MS and NMR experiments as described by *Mayr and Schieberle, 2012*. Deuterated solvents were obtained from Euriso-Top (Gif-Sur-Yvette, France). Solvents used for HPLC-MS/MS analysis were of LC-MS grade (Honeywell, Seelze, Germany); all other solvents were of HPLC grade (Merck Darmstadt, Germany). Water used for HPLC separation was purified by means of a Milli-Q water advantage A 10 water system (Millipore, Molsheim, France). [2H4]-Spermine, [2H2]-tyramine and [2H4]-spermidine were synthesized and purified as reported earlier (*Mayr and Schieberle, 2012*). Standard flyfood consisted of the following ingredients: agar, corn flour, soy flour, yeast, malt, molasses, nipagin, ethanol, phosphoric acid.

## Analysis of amines in fly food samples

Quantification of biogenic amines and polyamines in flyfood was performed by means of a stable isotope dilution LC-MS/MS method after derivatization as already reported by *Mayr and Schieberle, 2012*. Stock solutions. Stock solutions of the internal standards [13C3]-ethanolamine (52 µg/mL), [13C3,15N]-β-alanine (36.0 µg/mL), [2H4]-histamine (66.87 µg/mL), [2H4]-putrescine (28.86 µg/mL), [2H4]-spermine (91.56 µg/mL), [2H2]-tyramine (51 µg/mL) and [2H4]-spermidine (230 µg/mL) and [2H4]-phenylethylamine (109 µg/mL) were prepared in aqueous tri-chloro-acetic acid (10 %) and stored at 7 °C until use.

## Sample workup

Flyfood was frozen in liquid nitrogen and grounded in a mill (Moulinette, Moulinex, France). Aliquots (5 g) of each sample were spiked with an aliquot of the labled internal standards ([13C3, 15 N]β-alanine 50 µL, [2H4]-histamine 100 µL, [2H4]-putrescine 100 µL, [2H4]-spermine 50 µL, [2H2]-tyramine 100 µL, [2H4]-spermidine 50 µL, and [2H4]-phenyl-ethylamine 109 µL), thereafter, aqueous tri-chloro-acetic acid (10 %, 40 mL) was added, and vigorously stirred at room temperature. After an equilibration time of 30 min, the suspension was homogenated using an Ultraturrax (3 min, Jahnke and Kunkel, IKA-Labortechnik, Staufen im Breisgau, Germany), and ultrasonificated for another 10 min. The suspension obtained was centrifuged (10 min, 8000 rpm) and, finally, filtered (Schleicher & Schuell filter). The pH of the filtrate was adjusted to 10 with aqueous sodium hydroxide (1 M) and a solution of benzoyl chloride dissolved in acetonitrile (30 mL; 1 g/250 mL ACN) was added. After stirring for 2 hr at room temperature pH was adjusted to pH 2–3 using HCl (conc.). The benzamides were then extracted with dichloromethane (3x20 mL), and the organic phases were combined, dried over $Na_2SO_4$, and evaporated to dryness at 30 °C. The residue was dissolved in a mixture of acetonitrile and 0.1% aqueous formic acid (20/80, v/v) and filtered over a syringe filter (0.45 µm; Spartan 13/0.45 RC; Schleicher and Schuell, Dassel, Germany). The final filtrate was diluted with water (1/20, v/v). An aliquot (10 µL) of the prepared sample was injected into the HPLC-MS/MS system.

High Performance Liquid Chromatography-Triple Quadrupole Mass Spectrometry (HPLC-MS/MS). HPLC-MS/MS analysis was performed on a Surveyor high-performance chromatography system (Thermo Finnigan, Dreieich, Germany), equipped with a thermostated autosampler and a triple-quadrupol tandem mass spectrometer TQS Quantum Discovery (Thermo Electron, Dreieich, Germany). Temperature of the column compartment was set at 30 °C and autosampler temperature was 24 °C. After sample injection (10 µL), chromatography was carried out on a Synergy Fusion RP \ SI 80 Å column (150x2.0 mm id, 4 µm, Phenomenex, Aschaffenburg, Germany) using the following solvent gradient (0.2 mL/min) of acetonitrile (0.1% formic acid) as solvent A and formic acid (0.1% in water) as solvent B: 0 min, 0 % A; 1 min, 0 % A; 1.5 min, 35 % A; 20 min, 40 % A; 26 min, 50 % A; 27 min, 90 % A; 36 min, 90 % A; 37 min, 0 % A.; 52 min, 0 % A. The mass spectrometer was operated in the positive electrospray ionization (ESI+) mode, the spray needle voltage was set at 3.5 kV, the spray current at 5 µA, the temperature of the capillary was 300, the capillary voltage at 35 V. Nitrogen served as sheath and auxiliary gas, which was adjusted to 40 and 10 arbitary units. The collision cell was operated at a collision gas (argon) pressure of 0.13 Pa. Mass transitions of the pseudo molecular ions ([M+H]+) into specific product ions are summarized in *Mayr and Schieberle, 2012*. Calibration curves for the calculation of the response factors and linear ranges of the analytes were measured as described before by *Mayr and Schieberle, 2012*.

## Acknowledgements

We thank our project students C Biermeier, A Erb, D Mostert, V Statovskii, K Wald, R Raab and D Zapoglou for cross-validating T-maze experiments in independent experiments in single and multi-fly assays and extended research support for this project. We wish to acknowledge S Aimon for helping us with whole brain imaging and imaging data analysis. We especially thank Heidi Miller-Mommerskamp for technical support. Thanks also to A Bates, M Costa and G Jefferis for discussions regarding neurons of the lateral horn and access to EM tracing data. We acknowledge the Barnea, Jefferis and von Philipsborn labs for the provided fly lines. We are grateful to D Neumeier and S Kaviani-Nejad for technical assistance regarding the work on the polyamine concentration analysis in food. We also thank N Gompel, S Sachse, K Vogt and members of the Grunwald Kadow laboratory for comments on the manuscript and valuable feedback. This work was supported by the European Union (H2020, ERC starting grant 'FlyContext' to IGK) and the German research foundation (SFB870, A04 to IGK, and GR4310/5-1).

# Additional information

## Funding

| Funder | Grant reference number | Author |
| --- | --- | --- |
| European Research Council | ERC StG FlyContext | Anja B Friedrich |
| Deutsche Forschungsgemeinschaft | GR4310/5-1 | Ariane C Boehm Paul Bandow |
| Deutsche Forschungsgemeinschaft | CRC870 | KP Siju |
| Deutsche Forschungsgemeinschaft | A04 | KP Siju |

The funders had no role in study design, data collection and interpretation, or the decision to submit the work for publication.

## Author contributions

Ariane C Boehm, Conceptualization, Data curation, Formal analysis, Investigation, Supervision, Validation, Visualization, Writing – original draft, Writing – review and editing; Anja B Friedrich, Sydney Hunt, Data curation, Supervision, Visualization, Writing – review and editing; Paul Bandow, Validation, Supervision, Visualization, Methodology, Writing – review and editing; KP Siju, Jean Francois De Backer, Validation, Supervision, Visualization, Writing – review and editing; Julia Claussen, Marie Helen Link, Validation, Data curation, Supervision; Thomas F Hofmann, Conceptualization, Funding acquisition, Methodology; Corinna Dawid, Validation, Conceptualization, Data curation, Supervision, Methodology; Ilona C Grunwald Kadow, Formal analysis, Conceptualization, Funding acquisition, Supervision, Visualization, Writing – original draft, Project administration, Writing – review and editing

## Author ORCIDs

Jean Francois De Backer (ID) http://orcid.org/0000-0002-2861-9994
Marie Helen Link (ID) http://orcid.org/0000-0002-6065-2057
Ilona C Grunwald Kadow (ID) http://orcid.org/0000-0002-9085-4274

## Decision letter and Author response

Decision letter https://doi.org/10.7554/eLife.77643.sa1
Author response https://doi.org/10.7554/eLife.77643.sa2

# Additional files

## Supplementary files
• Transparent reporting form

## Data availability

Source Data files for all figures are available online: http://dx.doi.org/10.17632/5rz28jr8gc.2 Grunwald Kadow, Ilona (2022), "Boehm et al. (A dopamine-gated learning circuit underpins reproductive state-dependent odor preference in *Drosophila* females)", Mendeley Data, V2, https://doi.org/10.17632/5rz28jr8gc.2.

The following dataset was generated:

| Author(s) | Year | Dataset title | Dataset URL | Database and Identifier |
| --- | --- | --- | --- | --- |
| Boehm A, Bandow P, Siju KP, De Backer JF, Friedrich A, Hunt S, Link MH, Claussen J, Dawid C, Ilona C Grunwald Kadow | 2022 | A dopamine-gated learning circuit underpins reproductive state-dependent odor preference in *Drosophila* females, eLife | http://dx.doi.org/10.17632/5rz28jr8gc.2 | Mendeley Data, 10.17632/5rz28jr8gc.2 |

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
