## [Editor Report]

In this manuscript, the authors explore the circuit mechanism underlying mating-induced change of odor preference in *Drosophila*. Olfactory cues during mating induce a long-lasting increase in attraction to polyamines in female flies. The authors use a combination of neurogenetics, imaging, and behaviour to identify elements of the mushroom body and lateral horn circuitry involved in this behaviour. The importance of mushroom body plasticity in female postmating changes highlights a novel pathway for these changes and reveals the variety of mechanisms by which the brain can encode experience and adapt behavior, making this paper of interest to scientists within the field of reproductive behaviors and neuroscience of internal states.

---

## [Decision Letter]

**Decision letter after peer review:**

Thank you for submitting your article "A dopamine-gated learning circuit underpins reproductive state-dependent odor preference in *Drosophila* females" for consideration by *eLife*. Your article has been reviewed by 3 peer reviewers, including Sonia Sen as Reviewing Editor and Reviewer #1, and the evaluation has been overseen by K VijayRaghavan as the Senior Editor.

Essential revisions:

We appreciate the extensive body of work presented in this manuscript. All three reviewers were in agreement that the study is both of high quality and of interest to the community. There was also a general agreement that the paper took on parallel aspects of mating induced learning and recall of polyamine preference without resolving the circuit mechanism underlying either. Since blocking KCs is known to broadly affect olfactory coding (Wang et al., 2003), we recommend that the authors focus their analyses on the circuit elements required during mating (not recall). This would give the authors the opportunity to address some of the unresolved issues.

We summarise here a potential framework for the revised manuscript that we think will help focus the work. Listed alongside it are the supporting experiments we're suggesting. The new experiments are within.

1. cVA-sensitive OR67d-ORNs are required during mating for long-lasting polyamine preference change in mated females, although cVA alone isn't sufficient.

Figure 1A-B.

Figure 1E-G.

2. cVA activates PAM-β′1 and MBON-β′1.

Figure 1C-D (confirmatory data of Siju et al., for enhanced response in PAM-b'1)

Figure 5E.

– New experiments to test if PAM-β′1 can activate MBON-β′1 or test if cVA response of MBON-β′1 requires PAM-β′1.

3. Activation of PAM-β′1 or MBON-β′1 can change polyamine preference in virgin females.

Figure 3A (MB188B>dTrpA1 virgin data).

Figure 4B (MB057B>dTrpA1 virgin data).

– New experiments to test polyamine-specificity of results above by testing preference to ammonia.

– New experiments with MB025B>dTrpA1 to map sufficiency to PAM-β′1.

4. Output of MBON-β′1 is required during mating but not during test.

Figure 4B and S4B

– New experiment to reveal the requirement of DANs during mating by (a) blocking PAM-β′1 in SPR-/- background or (b) blocking PAM-β′1 with more potent effector such as TNT, Kir or reaper, or (c) blocking PAM-β′1 together with other DANs using broad driver line like 58E02-GAL4.

5. No plasticity in β′1 (MBON-β′1 likely affect polyamine pathway outside the MB)

Figure 5 and S5.

7. Identification of candidate LHON for polyamine representation and demonstration of their requirement during test.

Figure 6

– Potential new experiment to test polyamine response of LHAD1b2 in virgin and mated female.

In addition to this, please read the detailed reviews below and, wherever possible, do please try to address them.

*Reviewer #1 (Recommendations for the authors):*

–Related to the writing: I would urge the authors to revisit their writing. This is a dense paper and the manner in which it is currently written it makes hard for the reader to follow the data. There are a few things that I thought might be causing this. Could the authors please consider these, and any others, to make the data more accessible to the reader?

– The introduction doesn't equip the reader for what's to come. It doesn't cover many topics that are essential to the story in the manuscript. For example, it would have been nice to include material on mushroom body circuitry, the DA neurons and lateral horn circuitry. It would also have been nice to cover aspects of how these circuits encode associative learning in other contexts.

– Use of different names for the same neurons: The authors switch between names – sometimes referring to them by their Gal4 lines at others by their functional names.

– There are non-linear elements of the story telling that make it hard to follow any one thread. For example, the imaging data come up in three places – imaging of DANs in response to cVA, imaging of the whole brains, and finally the imaging of the MBONs to cVA. Have the authors considered addressing the circuit elements first, and then following how the relevant odours are represented in the represented in circuit elements?

– In general, it would help to have sentences with fewer clauses. At times these clauses are also unexpectedly ordered. A few of these can be a nice touch, but maybe there are a few too many here :

– Particularly in such dense data, it would really help to start each line of experiments with a motivation for why it was done.

– Related to the circuitry: The authors have identified various neurons that are involved in different aspects of the mating memory and recall behaviour (in both the MB and the LH). It wasn't clear to me how they think these neurons might function together within a circuit to bring about this behaviour. What's the link between these neurons? What's the link between the LH and MB induced behaviour? Aside from one experiment that stimulated the B1 DAN and monitored the activity in the MBONs, I see few examples of such 'epistatic' interrogation of circuitry. This left me feeling that I knew the parts, but didn't know how they tied up together. It would have been nice to see a few predictions of the circuitry tested as such here.

– Bidirectional behaviours: There are a couple of examples of bidirectional behaviours mediated by the same neurons in this manuscript. It was surprising to me, for example, that activating B1 DANs during mating interfered with the ability to mount a preference for polyamine, particularly because these neurons are active during mating (their imaging data). Is it possible that with thermogenetics, they are driving the neuron too hard? The P1 neuron mediates different behaviours when driven at different thresholds. Could such a phenomenon be occurring here? Have the authors tried to show this with another tool – csChimson, for example?

– cVA: I wasn't convinced that cVA was the relevant cue for this behaviour. Neither courtship nor cVA alone could reproduce the behavioural switch. This makes me think that it's likely a combination of pheromones and the physical act of mating that is driving the behaviour. So, I wonder how relevant it is to look for representation of cVA with the identified circuit elements -for example in the B1 DANs. Assessing the representation of the polyamine in a mating dependent way might be more relevant (and the authors have done this for many of their experiments).

Was there a reason why the authors decided to use an of 16 for the single fly experiments?

*Reviewer #2 (Recommendations for the authors):*

For a previous review I had asked some things, for example if exposure to cVA alone induces preference. I am quite happy to see these experiments now included in the new version of the paper, although it seems a coincidence that I am asked to review another time.

I think the paper has improved and it is also easier to read now.

Since I reviewed before, and my comments were seemingly already incorporated, I don't have so many new comments!

Suggestions:

1) line 151 following, Figure 1 C, D > I don't understand why these response to cVA experiments are shown here and how they fit with the natural flow of the story- just take them out? It is an interesting finding on its own but confuses the main story a bit. After all, the paper deals with the response to polyamine and how that changes after mating.

2) Figure 6 A: replace red green color scheme with one more suitable for red green blind.

*Reviewer #3 (Recommendations for the authors):*

Although the amount of data is impressive, it obscures main conclusions. Trim them to minimal set of data to support the main conclusions. For instance, data in Figure 3C and 3D are irrelevant to the model in Figure 7. Also some of data seem to be replication of previous reports and dispensable. For instance, Figure 1C-D are replication of Siju et al., 2020. ovoD1 mutant data was in Hussain et al., 2016. Otherwise, please clarify what is novel about these data. Along the same line, there is only weak link between data of Kenyon cells and LHON (Figure 2 and 6) and PAM-b'1/MBON-b'1.

[Editors’ note: further revisions were suggested prior to acceptance, as described below.]

Thank you for resubmitting your work entitled "A dopamine-gated learning circuit underpins reproductive state-dependent odor preference in *Drosophila* females" for further consideration by *eLife*. Your revised article has been evaluated by K VijayRaghavan (Senior Editor) and a Reviewing Editor.

The manuscript has been improved, and the reviewers are enthusiastic about its publication provided the following issues, as outlined below, are addressed.

The reviewers appreciate that the authors do not want to trim the data down as was recommended in the initial set of essential revisions. In this case, we are concerned about the specificity of the odour responses for many of the manipulations. However, to keep revisions to the minimum at this stage, our recommendations are the following:

1. Please move the recall experiments (Figure 3E, 3F, 4D, 4F , Figure 7—figure supplement 1) to supplementary data and discuss these in the "ideas and speculation" section. While doing so, do please explicitly state that it is possible that these manipulations might affect response to other odours, which has not been tested.

2. In the imaging data presented in Figure 6—figure supplement 1 we were concerned that the retinal minus control was actually activating DANs (perhaps through the contents of the cornmeal media). Could the authors please test that B'1 DANs are not actually active in their no retinal control by conducting the experiment in the absence of CsChrimson.

The detailed reviews are appended below.

*Reviewer #2 (Recommendations for the authors):*

In the revised version of the manuscript "A dopamine-gated learning circuit underpins reproductive state-dependent odor preference in *Drosophila* females », the authors addressed all comments and concerns raised by the reviewers. The manuscript still tells a complex story with some open ends that requires focus and concentration from the reader, but efforts have been made to make the data more accessible than in the first version.

It is helpful that the author inserted a short paragraph with an overview of the olfactory system to the introduction. Other adjustments to the text, e.g. in the discussion, also improved the manuscript.

Figure 1 is more streamlined now.

All requested experiments were carried out. Whenever the requested experiments did not give conclusive results due to supposed technical problems, this was explained in an appropriate way.

From my side, I accept and respect the author's argument that they would not like to completely omit all experiments addressing recall (as was suggested in order to simplify the manuscript).

The new results of LHAD1b2 changing its response to putrescine upon mating (Figure 7) is an interesting finding adding to the story.

*Reviewer #3 (Recommendations for the authors):*

As shown by the authors, a key feature of mating-induced change of odor preference is its specificity to polyamine. In this study, the authors presented many data that manipulation of some mushroom body cell types enhance or disrupt behavioral response to polyamine. However, such outcome of circuit manipulation does not necessarily mean that those cell types are responsible for induction or expression of polyamine-preference after mating. It could be because circuit manipulation recapitulated a situation after olfactory learning and affected response to arbitral odors including polyamine. Thus, in my opinion, the results of circuit manipulation are uninterpretable without control experiments to test polyamine-specificity (i.e. ammonia for virgin females and vinegar or other attractive odor for mated females).

For MBON-β′1, authors collected new data to show that its thermogenetic activation in virgin females induce long-lasting preference to polyamine but not to ammonia (Figure 4—figure supplement 1). Taken together with blocking experiments (Figure 4A), I'm convinced that MBON-β′1 plays an essential role in duction of mating-induced polyamine preference. I think the data about MBON-β′1 alone is worth publishing. I very much appreciate author's efforts to collect a large amount of data. However, showing them without control experiments about odor-specificity will be confusing to readers and even harmful to future investigation.

I suggested this experiment to establish functional connection between PAM-β′1 and MBON-β′1 either through direct DAN-to-MBON connection as depicted as an arrow in author's model in Figure 8, or through indirect connections. The outcome of DAN activation can be modulatory, but the data in Figure 6 and Figure 6—figure supplement 1A showed no evidence of mating-induced or DAN-activation-induced plasticity in the β′1 compartment. Therefore, I postulated that PAM-β′1 may directly activate MBON-β′1, as shown in other compartments (Takemura et al., 2017 *eLife*), to induce plasticity somewhere else.

The new imaging data in Figure 6—figure supplement 1B is hard to interpret without validation of retinal minus control. Without supplement of all-trans-retinal, cornmeal may contain enough precursor for biosynthesis of all-trans-retinal. Thus, to interpret the data, authors need to demonstrate that DANs are not activated by red light in retinal minus control or repeat the experiment without CsChrimson expression. I suspect that clear MBON-β′1 response to red light is due to activation of PAM-β′1 rather than visual response, especially when duration of activation is prolonged. Unlike γ dorsal Kenyon cells, there is no EM connectome data to support visual inputs to the α prime/β prime Kenyon cells.

If PAM-β′1 activation does not activate MBON-β′1, authors need to clarify, at lease in a working model, which synapses are modulated by PAM-β′1.

This new MB057B>dTrpA1 data about polyamine specificity of activation phenotype is a valuable addition. For MBON-β′1, there are a compelling set of data to conclude that MBON-β′1 is required during mating and its activity replace mating experience.

The data to demonstrate ployamine-specificity of the MB188B>dTrpA1 is still missing. I find it essential, especially because blocking PAM-DANs with MB188B>Shi during mating did not show any defect in mating-induced polyamine-preference. If 24-hour activation of DANs with MB188B>dTrpA1 change preference of other odors, it should be considered as artifact of long thermogenetic activation rather than recapitulation of mating experience.

I agree that Figure 2 "during mating" data is not expected and can be a strong result if authors could show polyamine-specificity of Kenon cell perturbation and evidence of mating-induced plasticity inside the mushroom body that is read out during odor test. I suggested to omit the Figure 2 data because the former requires additional experiments, and the latter is not supported by the presented data (Figure 6 and Figure 6—figure supplement 1).

To keep the Figure 2 data as a main figure, the vinegar (or other innately attractive odor in both virgins and mated females) should be used as a control odor to distinguish whether the defect of polyamine-preference by circuit-perturbation is due to impairment of forming/expressing mating-induced change or general defect in odor attraction as reported in Wang et al., 2003. As authors noted, a couple of papers claimed that KC's output is dispensable during training for olfactory conditioning. However, the timescale of manipulation is totally different. In the cited olfactory learning papers about short-term memories, temperature was elevated ~30min prior to the training and shifted lower to the permissive temperature immediately after training. In contrast, flies were kept at restrictive temperature for 24 hours in Figure 2 experiments, which likely include a long period after mating.

I suggest validating that mated MB010B>shi flies show intact vinegar attraction compared to control genotypes when temperature is elevated 24-hour period during/around mating or before/during test. Otherwise, I suggest omitting Figure 2 data.

As side notes about olfactory learning literatures, Krashes et al., 2008 Neuron showed that blocking α prime/β prime Kenyon cells impair memory, presumably because KCs are kept inactive for a while after temperature shift. Ichinose et al., 2015 *eLife* showed that output from the α/β KCs is necessary during training for long-term appetitive memory.

MB188>CsChrimson may not have such problem. I think that author's decision to tone down the conclusion is reasonable.

---

## [Author Response]

Essential revisions:We appreciate the extensive body of work presented in this manuscript. All three reviewers were in agreement that the study is both of high quality and of interest to the community. There was also a general agreement that the paper took on parallel aspects of mating induced learning and recall of polyamine preference without resolving the circuit mechanism underlying either. Since blocking KCs is known to broadly affect olfactory coding (Wang et al., 2003), we recommend that the authors focus their analyses on the circuit elements required during mating (not recall). This would give the authors the opportunity to address some of the unresolved issues.

We have added data and edited the text to address all criticisms and suggestions. As the reviewers have noted, we have collected a large amount of data over the course of several years. Like the reviewers, we had hoped that some data would significantly simplify the main conclusions and help us ‘distill’ a straight-forward mechanism. We have not been successful to identify the simple switch responsible for the change in behavior that we observe. In our opinion, given the importance of mating-induced behavior, it is not surprising that several parallel mechanisms ensure the proper behavioral transition from a virgin to a mated female. Therefore, we are somewhat reluctant to remove too much data or to streamline the paper too much. We don’t believe it would do justice to the biology. We hope that the reviewers will find this point of view acceptable. Nevertheless, we have edited the text to improve the reader’s experience and understanding of the data.

We summarise here a potential framework for the revised manuscript that we think will help focus the work. Listed alongside it are the supporting experiments we're suggesting. The new experiments are within – –.

Thank you very much for your constructive comments and helpful summary.

1. cVA-sensitive OR67d-ORNs are required during mating for long-lasting polyamine preference change in mated females, although cVA alone isn't sufficient.Figure 1A-B.Figure 1E-G.

As suggested, we have removed the imaging data that was shown in Figure 1C and D.

2. cVA activates PAM-β′1 and MBON-β′1.Figure 1C-D (confirmatory data of Siju et al., for enhanced response in PAM-b'1)Figure 5E.– New experiments to test if PAM-β′1 can activate MBON-β′1 or test if cVA response of MBON-β′1 requires PAM-β′1.

We have carried out the suggested experiment. Nevertheless, we would like to clarify an apparent misunderstanding. We do not propose that the PAM activates the MBON! We propose that the odor response of the PAM and the MBON originates primarily in the Kenyon cells. Connectomics data suggests that Kenyon cells form input synapses onto DANs and MBONs. In addition, we find several additional parallel routes from cVA OSNs to the PAMs as depicted in Figure 5 —figure supplement 1.

The synapse between PAM β′1 and MBON-β′1 is most likely a neuromodulatory (dopaminergic) and not an excitatory synapse.

Our experimental data support this assumption. Optogenetic activation of PAM-β′1 does not significantly activate MBON-β′1 as compared to controls. These data are included as Figure 6 —figure supplement 1.

We have also included several sentences in the text as well as a scheme of the circuit (see Figure 2 —figure supplement 1A) to clarify the relationship between PAM and MBON.

3. Activation of PAM-β′1 or MBON-β′1 can change polyamine preference in virgin femalesFigure 3A (MB188B>dTrpA1 virgin data).Figure 4B (MB057B>dTrpA1 virgin data).– New experiments to test polyamine-specificity of results above by testing preference to ammonia.– New experiments with MB025B>dTrpA1 to map sufficiency to PAM-β′1.

As suggested, we have carried out additional experiments:

– We have activated the MBON with line MB057B instead of mating and tested virgins for their preference for ammonia. We have not observed a change in preference upon activation of the MBON. This is in line with our experiments shown in Figure 1 —figure supplement 1A. We find no significant difference between mated and virgin females in their preference for ammonia. The new data are included in Figure 4 —figure supplement 1A.

– Regarding specificity, we have tested the preference of virgin and mated females for vinegar (1%) in the T-maze. Similar to ammonia, we find no difference in preference for vinegar between mated and virgin females (of note: the flies were fed to match the conditions we are using for the polyamine experiments). These data are included as Figure 1 —figure supplement 1B.

– We have tested the preference for polyamines upon activation of MB025B (instead of mating) to test the sufficiency of β′1. MB025B is expressed in PAM-β′1 ap and PAM-β′1 ap, while MB188B is expressed in PAM-β′1 ap (expression level 5), PAM-β′1 ap (expression level 5), as well as γ3 (expression level 3) and γ4 (expression level 2) albeit to a lower level. The results with line MB025B are difficult to interpret. For unknown reasons, the preference to polyamines in MB025B>TrpA1 females in single female T-maze experiments was already extremely high in virgins without the temperature shift instead of mating. We repeated the experiment at a lower concentration of polyamines (10mM) and found a similar result. The results are shown in Author response table 1. The differences, if any, between groups are not significant.

**Author response table 1. sa2table1:** 

100 mMPutrescine	not activated	activated
	trp MB025B 25C, 100 mM Putrescine	trp MB025B 32C, 100 mM Putrescine
percent choice	90	94.4
n(virgins) =	21	18
10 mMPutrescine		
	trp MB025B 18C, 10 mM Putrescine	trp MB025B 32C, 10 mM Putrescine
percent choice	58	71
n(virgins) =	36	35

We suspect that the genetic background of the line might be the reason for this high preference. We had seen a similar situation for a couple of other lines that we, for that reason, excluded from our initial behavioral screening.

Instead of showing these data in the manuscript, we have changed the text and figures to make clear that line MB188B includes not only β’1, but also γ3 and γ4 DANs.

4. Output of MBON-β′1 is required during mating but not during test.Figure 4B and S4B– New experiment to reveal the requirement of DANs during mating by (a) blocking PAM-β′1 in SPR-/- background or (b) blocking PAM-β′1 with more potent effector such as TNT, Kir or reaper, or (c) blocking PAM-β′1 together with other DANs using broad driver line like 58E02-GAL4.

We have carried out the experiment listed under (c). The experiment listed under (b) is difficult as TNT/Kir or reaper are not well suited to temporarily inhibit output during mating of the PAM neuron. Similarly for (a), we already know from our previously published results (Hussain, Ucpunar et al., PLoS Biology 2016) that SPR signaling is important and required. SPR-/- flies do not undergo the mating-related increase in polyamine attraction. Hence, inhibition of PAM-β’1 in that background is unlikely to reduce the lack of preference even further.

We have opted for experiment (c) and used the broader PAM-line MB42B which covers PAM-α1, PAM-β'1 apm, PAM-β’2 amp, PAM-β1, and PAM-γ3-γ5 to inhibit PAM output during mating. Similar to the results observed with inhibition of PAM-β′1, γ3,γ4 during mating, we have not observed a phenotype during test. These data are now including as Figure 3D.

Based on these data, we suggest -as also put forward by the reviewers- that the SPR pathway compensates and is sufficient to induce the observed mating-dependent increase in polyamine preference. We have also included this interpretation in the discussion.

5. No plasticity in β′1 (MBON-β′1 likely affect polyamine pathway outside the MB)Figure 5 and S5.

Please see below.

7. Identification of candidate LHON for polyamine representation and demonstration of their requirement during test.Figure 6– Potential new experiment to test polyamine response of LHAD1b2 in virgin and mated female.

As requested, we have imaged the response to polyamine of the LHAD1b2 neurons in virgin and mated female flies. The results are included as Figure 7. We find that the response of these LHONs is significantly different between polyamine and water only upon mating but not in virgin females. In virgins, LHAD1b2 neurons show the same the response to the water control as to the polyamine odor. Based on these results, we suggest that mating modulates the output of specific LHONs through changes in mushroom body output. These LHONs are important for the expression of the mating-induced increased preference to polyamines.

In addition to this, please read the detailed reviews below and, wherever possible, do please try to address them.

We have addressed all points below in detailed responses and, in some cases, by adding data. We hope that our changes to the text and figures and the new data will satisfy the reviewers and allow publication in *eLife*.

Reviewer #1 (Recommendations for the authors):–Related to the writing: I would urge the authors to revisit their writing. This is a dense paper and the manner in which it is currently written it makes hard for the reader to follow the data. There are a few things that I thought might be causing this. Could the authors please consider these, and any others, to make the data more accessible to the reader?

We thank you for the suggestions below. We have made several edits to the text to streamline data presentation.

– The introduction doesn't equip the reader for what's to come. It doesn't cover many topics that are essential to the story in the manuscript. For example, it would have been nice to include material on mushroom body circuitry, the DA neurons and lateral horn circuitry. It would also have been nice to cover aspects of how these circuits encode associative learning in other contexts.

We have added a paragraph to the introduction to equip the reader for the results as suggested by the reviewer. We have also introduced an additional schematic figure as Figure 2A depicting the connections between neurons of the MB and LH.

– Use of different names for the same neurons: The authors switch between names – sometimes referring to them by their Gal4 lines at others by their functional names.

We agree with this reviewer that the use of Gal4 and their names/innervation patterns makes it somewhat complicated to follow. Unfortunately, we don’t see an easy solution to this problem. First, several MB-lines cover many neurons and using the names of the neurons in the text would make the text even less readable. Second, using only the MB-line names would obscure which neurons are covered and therefore make it more complicated for people not familiar with the MBline collection by Aso et al. Nevertheless, we tried to reduce the incidents where we’ve used both as much as possible.

– There are non-linear elements of the story telling that make it hard to follow any one thread. For example, the imaging data come up in three places – imaging of DANs in response to cVA, imaging of the whole brains, and finally the imaging of the MBONs to cVA. Have the authors considered addressing the circuit elements first, and then following how the relevant odours are represented in the represented in circuit elements?

We followed this suggestion. We have removed the imaging data in Figure 1 as similar findings were published by us previously in Siju et al., Current Biology 2020. In addition, we have moved the LHON behavioral data (former Figure 6) before the MBON imaging figure (former Figure 5). In the revised manuscript, Figure 1-5 include only behavioral and anatomical data while Figure 6 and 7 contain in vivo imaging results.

– In general, it would help to have sentences with fewer clauses. At times these clauses are also unexpectedly ordered. A few of these can be a nice touch, but maybe there are a few too many here :

We have reduced the complexity of the sentences. Thank you for this suggestion.

– Particularly in such dense data, it would really help to start each line of experiments with a motivation for why it was done.

We have added this to the text.

– Related to the circuitry: The authors have identified various neurons that are involved in different aspects of the mating memory and recall behaviour (in both the MB and the LH). It wasn't clear to me how they think these neurons might function together within a circuit to bring about this behaviour. What's the link between these neurons? What's the link between the LH and MB induced behaviour? Aside from one experiment that stimulated the B1 DAN and monitored the activity in the MBONs, I see few examples of such 'epistatic' interrogation of circuitry. This left me feeling that I knew the parts, but didn't know how they tied up together. It would have been nice to see a few predictions of the circuitry tested as such here.

We agree that this would be helpful. To this end, we have now included the experiment suggested above to test whether PAM-β’1 neurons are capable of activating MBON-β’1 (Figure 6 —figure supplement 1). In this particular case, we did not mean to suggest that PAMs would activate MBONs. Instead, we assume a neuromodulatory relationship between DANs and MBONs. We had hoped that former Figure 7 (now Figure 8) would help to piece the circuit together. As mentioned above, we have now introduced another schematic to clarify the synaptic connections of the MB (Figure 2A). In virgins, our data suggest that the MB (i.e. β’2 region) prevents high attraction to polyamines. During mating, in part through cVA and MBON-β’1, this effect of the β’2 compartment is suppressed. These neurons, as indicated in Figure 4 —figure supplement 1, are indirectly connected. Thus, mating changes the balance in the MB circuit, which in turn leads to, possibly through MBON-α2sc (Figure 7 —figure supplement 1), a change in the LHON response to polyamine (Figure 7).

We absolutely agree that epistasis experiments would be very useful and we plan to do them in the future.

– Bidirectional behaviours: There are a couple of examples of bidirectional behaviours mediated by the same neurons in this manuscript. It was surprising to me, for example, that activating B1 DANs during mating interfered with the ability to mount a preference for polyamine, particularly because these neurons are active during mating (their imaging data). Is it possible that with thermogenetics, they are driving the neuron too hard? The P1 neuron mediates different behaviours when driven at different thresholds. Could such a phenomenon be occurring here? Have the authors tried to show this with another tool – csChimson, for example?

This is possible. We have repeated the same experiment with csChrimson to induce preference in virgins for the imaging experiments shown in Figure 5 —figure supplement 1. Strong activation over 24h with pulsed light led to the same phenotype as activation with TrpA1. We have not systematically tested different light intensities, but this would be a good idea. Unfortunately, as my lab has moved during the week of July 11. We, therefore, had to prioritize the experiments listed above to finish whatever was possible before the move.

– cVA: I wasn't convinced that cVA was the relevant cue for this behaviour. Neither courtship nor cVA alone could reproduce the behavioural switch. This makes me think that it's likely a combination of pheromones and the physical act of mating that is driving the behaviour. So, I wonder how relevant it is to look for representation of cVA with the identified circuit elements -for example in the B1 DANs. Assessing the representation of the polyamine in a mating dependent way might be more relevant (and the authors have done this for many of their experiments).

We completely agree that cVA is just one of the cues needed for the switch. Our data in Figure 1CE, however, show that inhibition of cVA detecting neurons during mating reduces later polyamine preference. In our opinion, this indicates an important function of cVA in the mating induced change.

Was there a reason why the authors decided to use an of 16 for the single fly experiments?

Yes, there are two main reasons: 1. 16 single experiments for multiple groups could be carried out in one to two consecutive days, which is what we aimed for given the sensitivity of state-dependent behavior to all kinds of confounds. 2. 16 flies were sufficient to observe strong phenotypes. Given the vast number of neurons in and around the mushroom body, we decided to focus on the neurons that resulted in a strong change rather than milder phenotypes.

Reviewer #2 (Recommendations for the authors):Suggestions:1) line 151 following, Figure 1 C, D > I don't understand why these response to cVA experiments are shown here and how they fit with the natural flow of the story- just take them out? It is an interesting finding on its own but confuses the main story a bit. After all, the paper deals with the response to polyamine and how that changes after mating.

We have removed these data from Figure 1.

2) Figure 6 A: replace red green color scheme with one more suitable for red green blind.

We have changed the colors in this panel. Thank you for alerting us to this problem.

Reviewer #3 (Recommendations for the authors):Although the amount of data is impressive, it obscures main conclusions. Trim them to minimal set of data to support the main conclusions. For instance, data in Figure 3C and 3D are irrelevant to the model in Figure 7. Also some of data seem to be replication of previous reports and dispensable. For instance, Figure 1C-D are replication of Siju et al., 2020. ovoD1 mutant data was in Hussain et al., 2016. Otherwise, please clarify what is novel about these data. Along the same line, there is only weak link between data of Kenyon cells and LHON (Figure 2 and 6) and PAM-b'1/MBON-b'1.

We agree and, as written above, we had hoped that we would eventually obtain a very clear cut and simple result to explain our observation. We have removed Figure 1 C and D. We have also removed the *ovoD1* mutant data, which had only been included following a previous reviewer comment. We believe that the results in Figure 2 are very important, because they show that KC output is required during mating. KCs are a major part of the olfactory pathway and provide input to DANs and MBONs. Regarding the role of the LHONs, we now show that these neurons change their response to putrescine after mating (Figure 7). In addition, we show that MBON-α2sc, which has been previously shown to connect the mushroom body to the lateral horn, is important for the expression of the behavior (Figure 7 —figure supplement 1). [Editors’ note: what follows is the authors’ response to the second round of review.]

Reviewer #2 (Recommendations for the authors):In the revised version of the manuscript "A dopamine-gated learning circuit underpins reproductive state-dependent odor preference in *Drosophila* females », the authors addressed all comments and concerns raised by the reviewers. The manuscript still tells a complex story with some open ends that requires focus and concentration from the reader, but efforts have been made to make the data more accessible than in the first version.

We have now moved all ‘recall’ data into the supplementary figures in hopes of making the manuscript more straight forward.

It is helpful that the author inserted a short paragraph with an overview of the olfactory system to the introduction. Other adjustments to the text, e.g. in the discussion, also improved the manuscript.Figure 1 is more streamlined now.All requested experiments were carried out. Whenever the requested experiments did not give conclusive results due to supposed technical problems, this was explained in an appropriate way.From my side, I accept and respect the author's argument that they would not like to completely omit all experiments addressing recall (as was suggested in order to simplify the manuscript).The new results of LHAD1b2 changing its response to putrescine upon mating (Figure 7) is an interesting finding adding to the story.

We thank this reviewer very much for their support. We have modified the schematic in figure 3 and in all figures where virgin and mated data were shown by inserting a virgin sign into the schematic.

Reviewer #3 (Recommendations for the authors):As shown by the authors, a key feature of mating-induced change of odor preference is its specificity to polyamine. In this study, the authors presented many data that manipulation of some mushroom body cell types enhance or disrupt behavioral response to polyamine. However, such outcome of circuit manipulation does not necessarily mean that those cell types are responsible for induction or expression of polyamine-preference after mating. It could be because circuit manipulation recapitulated a situation after olfactory learning and affected response to arbitral odors including polyamine. Thus, in my opinion, the results of circuit manipulation are uninterpretable without control experiments to test polyamine-specificity (i.e. ammonia for virgin females and vinegar or other attractive odor for mated females).

We respectfully disagree. First, we have never claimed that our findings apply exclusively to polyamines. We have added additional data regarding specificity because Reviewer 3 requested it. At this point, we can only say that mating does not change the response to any odor but appears to be restricted to some, possibly physiologically important, odors.

Polyamines are a model for the behavior we observe. Given that neither MBONs, DANs nor LH neurons are specific for any single odorant (e.g. Dolan et al., 2019; Hige et al., 2015), we think it is quite unlikely that this mechanism only applies to polyamines. In fact, our data showing that the presence of polyamines during mating is not required for the switch in behavior shows that this is NOT a form of associative learning. It is long-term plasticity that appears to involve components of the mushroom body as well as the lateral horn.

Nevertheless, even in associative memory paradigms, animals including flies can generalize to other odors that were not present during learning.

We have added and modified the text throughout to make this clearer. We do not claim that this mechanism only applies to polyamines. However, in disagreement with Reviewer 3, we do not see a problem with this interpretation. Mating induces a change that affects many aspects of behavior.

Polyamines, likely for physiological reasons, are indicators for beneficial environments upon mating. Ammonia is not. However, there might be other odors that signal a similar beneficial condition.

For MBON-β′1, authors collected new data to show that its thermogenetic activation in virgin females induce long-lasting preference to polyamine but not to ammonia (Figure 4—figure supplement 1). Taken together with blocking experiments (Figure 4A), I'm convinced that MBON-β′1 plays an essential role in duction of mating-induced polyamine preference. I think the data about MBON-β′1 alone is worth publishing. I very much appreciate author's efforts to collect a large amount of data. However, showing them without control experiments about odor-specificity will be confusing to readers and even harmful to future investigation.

Please see above. Odor specificity to just one odor, i.e. polyamines, is unlikely given the makeup of the neurons and circuits involved. This mechanism might very well also extend to other, similarly indicative, odors.

I suggested this experiment to establish functional connection between PAM-β′1 and MBON-β′1 either through direct DAN-to-MBON connection as depicted as an arrow in author's model in Figure 8, or through indirect connections. The outcome of DAN activation can be modulatory, but the data in Figure 6 and Figure 6—figure supplement 1A showed no evidence of mating-induced or DAN-activation-induced plasticity in the β′1 compartment. Therefore, I postulated that PAM-β′1 may directly activate MBON-β′1, as shown in other compartments (Takemura et al., 2017 eLife), to induce plasticity somewhere else.The new imaging data in Figure 6—figure supplement 1B is hard to interpret without validation of retinal minus control. Without supplement of all-trans-retinal, cornmeal may contain enough precursor for biosynthesis of all-trans-retinal. Thus, to interpret the data, authors need to demonstrate that DANs are not activated by red light in retinal minus control or repeat the experiment without CsChrimson expression. I suspect that clear MBON-β′1 response to red light is due to activation of PAM-β′1 rather than visual response, especially when duration of activation is prolonged. Unlike γ dorsal Kenyon cells, there is no EM connectome data to support visual inputs to the α prime/β prime Kenyon cells.

We agree that it is possible that the food source included sufficient retinal to induce Chrimson activation. Unfortunately, we simply did not have enough time to include additional controls before our laboratory move. At this moment, we do not have a functional 2-P microscope (we have ordered one but it will only arrive in 4-6 months unfortunately) to carry out additional experiments. We have therefore toned down our claims regarding this experiment in the text with the note that additional experiments would be needed to confirm the results.

If PAM-β′1 activation does not activate MBON-β′1, authors need to clarify, at lease in a working model, which synapses are modulated by PAM-β′1.

We still believe it is possible that PAM-β′1 modulates the response of MBON-β′1. In addition, it is also possible, as suggested by Reviewer 3, that the DAN directly activates the MBON. We have now included both scenarios in the text.

Why do we not see plasticity at the MBON? We speculate in the text that this is due to finding the correct time window. This modulation might be very transient and eventually manifest in long-term changes in other neurons such as the LHON AD1b2 as we have shown.

We now state explicitly that a direct activation of the MBON through the PAM is a possible scenario.

This new MB057B>dTrpA1 data about polyamine specificity of activation phenotype is a valuable addition. For MBON-β′1, there are a compelling set of data to conclude that MBON-β′1 is required during mating and its activity replace mating experience.The data to demonstrate ployamine-specificity of the MB188B>dTrpA1 is still missing. I find it essential, especially because blocking PAM-DANs with MB188B>Shi during mating did not show any defect in mating-induced polyamine-preference. If 24-hour activation of DANs with MB188B>dTrpA1 change preference of other odors, it should be considered as artifact of long thermogenetic activation rather than recapitulation of mating experience.

As mentioned above, we do not believe that the described mechanism only applies to polyamines. We have shown however that these DANs are activated by cVA (Siju et al., 2020), which shows that mating-related information is detected by these neurons. It is conceivable that this PAM is not specific to cVA or mating, but instead responds to additional cues or conditions. For instance, we show in Siju et al., 2020 that these neurons are also modulated by hunger. Thus, this type of PAM might be more generally involved in internal state-dependent behavior.

In 2015, we have shown that PAM- β′2 is involved in the resolution of sensory conflict dependent on hunger (Lewis et al., 2015). In parallel, other groups have shown that this neuron is also involved in associative learning, arousal from sleep etc (e.g. Owald et al., 2015; French et al., 2021). Similar scenarios are observed in the mammalian amygdala (e.g. Douglass et al., 2017). In line with this, we do not postulate that the β′1 region is only important for mating state-dependent behavior to a single type of odor.

We have tried to make this clearer in the text.

I agree that Figure 2 "during mating" data is not expected and can be a strong result if authors could show polyamine-specificity of Kenon cell perturbation and evidence of mating-induced plasticity inside the mushroom body that is read out during odor test. I suggested to omit the Figure 2 data because the former requires additional experiments, and the latter is not supported by the presented data (Figure 6 and Figure 6—figure supplement 1).

We respectfully disagree. In our view, the interesting finding is that KC output is needed during mating to lastingly induce a change in odor preference.

To keep the Figure 2 data as a main figure, the vinegar (or other innately attractive odor in both virgins and mated females) should be used as a control odor to distinguish whether the defect of polyamine-preference by circuit-perturbation is due to impairment of forming/expressing mating-induced change or general defect in odor attraction as reported in Wang et al., 2003. As authors noted, a couple of papers claimed that KC's output is dispensable during training for olfactory conditioning. However, the timescale of manipulation is totally different. In the cited olfactory learning papers about short-term memories, temperature was elevated ~30min prior to the training and shifted lower to the permissive temperature immediately after training. In contrast, flies were kept at restrictive temperature for 24 hours in Figure 2 experiments, which likely include a long period after mating.I suggest validating that mated MB010B>shi flies show intact vinegar attraction compared to control genotypes when temperature is elevated 24-hour period during/around mating or before/during test. Otherwise, I suggest omitting Figure 2 data.

Again, we respectfully disagree. We maintain that the finding in Figure 2 is very important. Whether it is specific for polyamines or not is not relevant in our view. The important finding is that KC output is required during mating so that the change in mating-induced odor preference can happen. Importantly, 24h inhibition of KC output before or after mating did not have the same effect. This strongly suggests that something specific to mating and involving the KC/MB neurons is taking place. This, in our view, is the relevant finding. If we were to see the same with another odorant, this would not change anything regarding our conclusions as we do not see odor specificity as the important point.

As side notes about olfactory learning literatures, Krashes et al., 2008 Neuron showed that blocking α prime/β prime Kenyon cells impair memory, presumably because KCs are kept inactive for a while after temperature shift. Ichinose et al., 2015 eLife showed that output from the α/β KCs is necessary during training for long-term appetitive memory.

We already know from previous work (for instance Lewis et al., 2015) that KCs are involved in innate odor responses. For the same reasons as mentioned above, we don’t see this as a problem. In our opinion, no circuit in any brain also subserves a single function (perhaps with the exception of MN connecting to a specific muscle).10.17632/5rz28jr8gc.2